# Don't Forget Why You Started: Tackling Dual Forgetting in Vision-Language Continual Learning

**Borui Kang** [† 1]  **Jinrui Gu** [† 1]  **Tao Feng** [2]  **Qi Fan** [1]  **Yinghuan Shi** [1]  **Lei Wang** [3]  **Wenbin Li** [* 1]  **Yang Gao** [1]

## Abstract

Continual learning of Vision-Language Model (VLM) aims to empower models with new expertise without compromising zero-shot capabilities. However, this pursuit faces a critical "dual-forgetting" challenge: catastrophic forgetting of newly acquired classes (Incremental Knowledge Forgetting, IKF) and erosion of foundational zero-shot capabilities (Pre-trained Knowledge Forgetting, PKF). Existing evaluations often ignore PKF or assess it via confounded protocols where *positive transfer* on semantically similar domains creates an *illusion of retention*, masking severe foundational degradation. To address this, we propose the *Dual-Forgetting-Aware Class-Incremental Learning (DFA-CIL)* framework and the *Similarity-Calibrated Retention (SCR)* metric. Unlike standard averaging, SCR uses the frozen pre-trained feature space to inversely weight performance by semantic similarity, mitigating confounding gains to stress-test foundational stability. Building on this, we propose *DFA-MoE, a functionally heterogeneous Parameter-Efficient Fine-Tuning (PEFT) method*. DFA-MoE decouples optimization objectives by assigning a momentum-enhanced contrastive expert for feature alignment, and separate plasticity experts combining classification with auxiliary contrastive learning to adapt to new tasks while retaining historical knowledge. Extensive experiments show that our framework reveals the hidden fragility of existing methods and achieves a state-of-the-art balance in preserving both incremental and pre-trained knowledge. Our code is available at https://github.com/RL-MIND/DFA-MoE.

[†]Equal contribution  [*]Corresponding author [1]State Key Laboratory of Novel Software Technology, Nanjing University, Nanjing 210023, China [2]Tsinghua University, China [3]School of Computing and Information Technology, University of Wollongong, Australia. Correspondence to: Wenbin Li <liwenbin@nju.edu.cn>.

*Proceedings of the 43rd International Conference on Machine Learning*, Seoul, South Korea. PMLR 306, 2026. Copyright 2026 by the author(s).

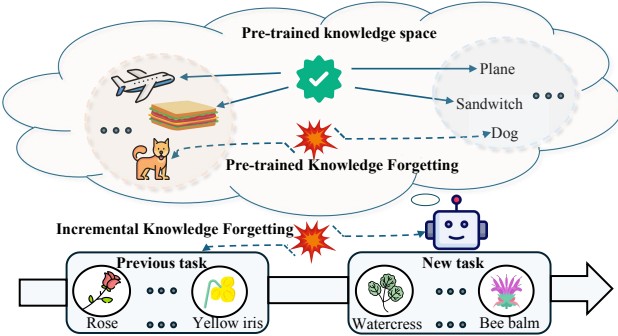

*Figure 1.* The dual-forgetting in VLM continual learning. When fine-tuning on a *New Task* (*e.g.*, "Watercress"), the model faces two distinct risks: (1) IKF, losing previously learned specific classes (*e.g.*, "Rose"); and (2) PKF, where the foundational alignment required to recognize general concepts (*e.g.*, "Dog") is eroded.

## 1. Introduction

The zero-shot generalization of Vision-Language Model (VLM) such as CLIP (Radford et al., 2021; Liu et al., 2025) is often compromised by traditional continual learning, which warps the feature space to fit specific tasks (De Lange et al., 2021; Wang et al., 2024). This sacrifice of *foundational stability*—the integrity of the original semantic geometry—triggers the *dual-forgetting problem* (Figure 1). As fragile multi-modal alignments fracture, the model suffers two distinct losses: *Incremental Knowledge Forgetting (IKF)* of previously learned classes (McCloskey & Cohen, 1989), and *Pre-trained Knowledge Forgetting (PKF)*, the erosion of VLM's foundational zero-shot capabilities (Zheng et al., 2023; Dong et al., 2025).

While prior research has made progress on IKF, the evaluation of PKF remains insufficiently rigorous. Class-Incremental Learning (CIL) methods (Van de Ven et al., 2022; Van de Ven & Tolias, 2019; Yu et al., 2025b) typically ignore PKF entirely. Conversely, frameworks (Zheng et al., 2023) that do acknowledge PKF often rely on naive average performance metrics that fail to account for the semantic correlation between incremental tasks and evaluation domains. A critical ambiguity arises here: *In continual learning, does a high zero-shot score indicate genuine retention of foundational knowledge, or merely positive transfer?*

For instance, fine-tuning on a concept like "Rose" naturally boosts performance on the semantically proximate "Tulip" via positive transfer. While beneficial for application, this transfer acts as a confounding factor during evaluation. Within the continual learning paradigm, *observed performance* inherently conflates *foundational stability* with *transfer effects*. On similar domains, strong transfer effects compensate for the erosion of foundational stability, masking the structural distortion of the original feature space. Consequently, a model may appear robust while its ability to recognize dissimilar concepts—where transfer is negligible—has degraded, creating a deceptive *illusion of retention*.

To rigorously diagnose this, we propose the *Dual-Forgetting-Aware Class-Incremental Learning (DFA-CIL)* framework to quantify IKF and PKF. To obtain a more reliable estimate of pre-trained knowledge retention under positive-transfer confounding, we introduce the *Similarity-Calibrated Retention (SCR)* metric. By applying *similarity-inverse weighting*, SCR recalibrates evaluation: it strategically down-weights high-similarity domains to disentangle the confounding effects of positive transfer, while prioritizing low-similarity domains as faithful proxies for the pre-trained foundation. This prevents local generalization from masking global degradation, revealing the VLM's intrinsic foundational stability during continual learning.

Building on these insights, we propose *DFA-MoE, a functionally heterogeneous Parameter-Efficient Fine-Tuning (PEFT) method*. Unlike homogeneous Mixture-of-Experts (MoE) (Jacobs et al., 1991) where all experts share a unified objective, DFA-MoE enforces *distinct functional pathways* to resolve the dual-forgetting conflict. We decouple the objectives: an *Alignment Pathway* is governed by a momentum-enhanced contrastive objective to anchor the VLM's foundational alignment, while a *Plasticity Pathway* employs a dual-objective strategy: optimizing for new class discrimination while leveraging historical contrast to prevent IKF. This design resolves the stability-plasticity tension.

Our main contributions are summarized as follows:

- We identify the dual-forgetting problem (IKF & PKF) in VLM continual learning and propose the DFA-CIL framework to establish a rigorous evaluation protocol.

- We introduce the SCR metric, which utilizes similarity-based inverse weighting to disentangle genuine foundational retention from the confounding effects of positive transfer in CIL.

- We propose DFA-MoE, a functionally heterogeneous PEFT framework. By architecturally decoupling the optimization objectives and introducing a composite loss for plasticity experts, our method mitigates both PKF and IKF, achieving a state-of-the-art balance.

## 2. Related Work

**VLM-based Class-Incremental Learning.** Recent studies have revealed that VLM such as CLIP (Radford et al., 2021) are inherently effective continual learners (Thengane et al., 2022; Ding et al., 2022; Huang et al., 2025). Leveraging this adaptability, researchers have explored various strategies. One line of work adapts unimodal techniques to the multi-modal context. This includes methods such as applying attention distillation (Lu et al., 2024) or designing cross-modality regularization strategies (Cui et al., 2024; Yu et al., 2024b). A more prevailing direction is PEFT (Houlsby et al., 2019). Within this paradigm, architectural approaches like MoE4Adapter (Yu et al., 2024a) use the Mixture-of-Experts architecture to dynamically allocate parameters, while optimization-focused methods like DMNSP (Kang et al., 2025) employ null space projection for more refined gradient control. Other PEFT strategies, including prompt-based learning (Wang et al., 2022) and various adapter designs (Gao et al., 2024; Sun et al., 2025), also strive to balance adaptation with knowledge retention. Distinct from these parameter-isolation strategies, another major and highly effective approach is synthetic replay. Methods in this category such as GIFT (Wu et al., 2025) leverage diffusion models to generate data from past tasks, enabling knowledge distillation from the VLM of the previous stage.

However, these methods suffer from a common oversight: they focus almost exclusively on mitigating IKF, effectively treating the pre-trained backbone merely as a feature extractor for new tasks. This narrow focus neglects PKF, failing to monitor the erosion of the VLM's original vision-language alignment. Consequently, these approaches risk sacrificing foundational stability for incremental plasticity, reducing a general-purpose agent to a specialized classifier.

**Multi-domain Task-Incremental Learning.** The MTIL framework, introduced by ZSCL (Zheng et al., 2023), assesses continual learning via zero-shot evaluation on upcoming tasks. Subsequent work uses adapter-based (Zhou et al., 2025; Luo et al., 2025; Lu et al., 2026) or prompt-based (Fu et al., 2025; Wang & Chen, 2025; Tang et al., 2024) PEFT strategies to isolate task-specific knowledge. However, MTIL has a critical evaluation flaw: as detailed in Section 3.2, it overlooks semantic overlap between learned tasks and evaluation domains.

Consequently, the metrics of MTIL often conflate positive transfer from the current task with genuine retention of the pre-trained foundation. This creates an illusion of stability, where inflated scores on similar domains mask knowledge erosion on dissimilar ones. Furthermore, MTIL's reliance on task identities and a shrinking evaluation scope limits its validity for measuring foundational stability in realistic, task-id-free settings.

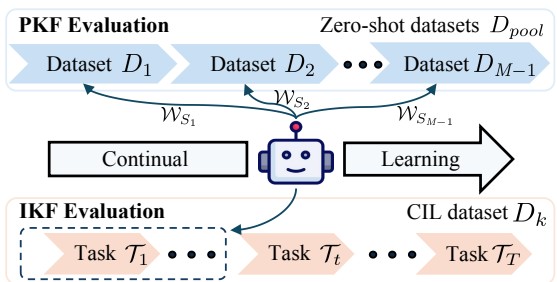

*Figure 2.* The proposed overall DFA-CIL framework for measuring IKF and PKF. The model continually learns a sequence of tasks, $\{\mathcal{T}_1, \ldots, \mathcal{T}_T\}$, sourced from the CIL dataset $D_k$ (bottom). After learning task $\mathcal{T}_t$, its IKF is evaluated on previous tasks $\{\mathcal{T}_1, \ldots, \mathcal{T}_{t-1}\}$, while its PKF is assessed on the evaluation pool $D_{pool}$ (top).

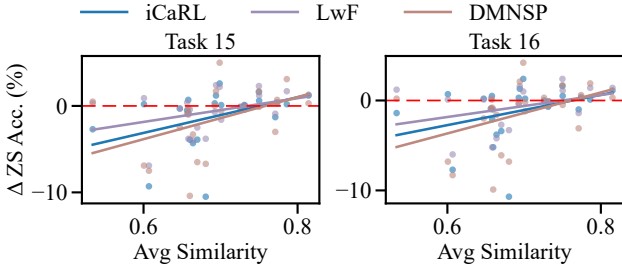

*Figure 3.* Visualizing the confounding effect of positive transfer on zero-shot evaluation. We plot the change in zero-shot accuracy ($\Delta$ZS Acc.) for each evaluation dataset (represented as each point) as a function of its similarity to the training tasks (OxfordFlower CIL, tasks 15-16). See Supplementary Material E for a more extensive analysis.

## 3. The proposed DFA-CIL Framework

We design the DFA-CIL framework, the first principled protocol to evaluate both IKF and PKF in VLM continual learning. This framework incorporates a multi-modal similarity analysis to disentangle the confounding influence of positive transfer, ensuring that foundational stability is assessed solely based on the preservation of original capabilities rather than new adaptations.

### 3.1. Composition in DFA-CIL Framework

The DFA-CIL framework is built upon a diverse collection of $M$ datasets, $\{D_1, \ldots, D_M\}$. This collection is drawn from the original zero-shot evaluation sets of CLIP (Radford et al., 2021) to act as a rigorous testbed for generalizability. As shown in Figure 2, each experiment follows a partitioning scheme: a single dataset $D_k$ is selected from the collection to serve as the CIL source dataset, while the remaining $M-1$ datasets are aggregated to form the zero-shot evaluation pool ($D_{pool}$). $D_k$ is then partitioned into a sequence of incremental tasks, $\{\mathcal{T}_1, \ldots, \mathcal{T}_T\}$, which are learned under a task-id-free protocol. Here, task-id-free means that no task identifier is available at inference, so the model must make predictions over the unified set of all seen classes. After each task $\mathcal{T}_t$ is completed, the model's performance is measured on two fronts: its IKF is measured on the test sets of all previously seen classes from $D_k$, while its PKF is measured on each dataset of the zero-shot evaluation dataset pool $D_{pool}$. Crucially, we refine this PKF evaluation by applying a set of similarity-inverse weights, denoted by $\{W_{S_1}, \ldots, W_{S_{M-1}}\}$. Each weight $W_{S_i}$ is applied to its corresponding dataset in $D_{pool}$.

### 3.2. Evaluation in DFA-CIL

**Incremental Knowledge Evaluation.** Following the literature (Wang et al., 2024; Masana et al., 2022), the retention of incremental knowledge is measured using two standard

CIL metrics. After each task, we compute: (1) the Last Accuracy across the test sets of all classes seen so far ($\cup_{i=1}^{t} C_i$), reflecting overall performance; and (2) Backward Transfer (BWT), which quantifies the impact of learning the current task ($\mathcal{T}_t$) on the performance of previous tasks, $\mathcal{T}_1, \ldots, \mathcal{T}_{t-1}$. A higher BWT score indicates less forgetting.

**Disentangling Retention from Positive Transfer.** The validity of zero-shot evaluation for PKF relies on the assumption that the target domain tests the original pre-trained knowledge. However, as visualized in Figure 3, we observe a significant positive correlation between the semantic similarity of evaluation domains and their performance retention. This indicates that high scores on proximate domains are substantially driven by *positive transfer* from the current task rather than genuine stability. While such transfer is beneficial for local adaptation, it acts as a *confounding factor* that masks foundational erosion, creating an *illusion of retention*. Crucially, this masking effect obscures a critical failure mode: the VLM is silently sacrificing its generic zero-shot versatility to accommodate recent tasks, degenerating from a general-purpose agent into a narrow specialist—a direct violation of the core objective of VLM continual learning. Consequently, the true fragility of the pre-trained weights is only exposed on disjoint, dissimilar domains where the model cannot rely on recent transfer. Therefore, we down-weight similar domains not to discard valid performance, but to disentangle uninformative signals and rigorously diagnose this foundational decay.

**The Similarity-Calibrated Retention Metric.** To implement this rigorous assessment, we propose the Similarity-Calibrated Retention (SCR) metric. Crucially, to ensure a stable and fair comparison across different methods, all similarity computations are performed using the original, frozen CLIP backbone ($\mathcal{M}_o$). This establishes a static, objective ruler for semantic proximity, independent of the model's drifting feature space during training.

Specifically, for the current training task $D_t$ and a zero-shot evaluation dataset $D_e$, we first extract their feature representations using $\mathcal{M}_o$. For each class $i$, we construct a multimodal prototype $\mathbf{r}^i = [\mathbf{v}^i; \mathbf{t}^i] \in \mathbb{R}^{p+q}$ by concatenating its $p$-dimensional visual features $\mathbf{v}^i$ and $q$-dimensional textual features $\mathbf{t}^i$, where both feature vectors are independently L2-normalized prior to concatenation to ensure balanced modal contributions. The semantic gap is then quantified by the Directed Chamfer Distance (Fan et al., 2017) (see Supplementary Material E for a detailed discussion on the rationale). For the current training task $D_t$ (size $m$) at incremental step $t$ and a target evaluation dataset $D_e$ (size $n$), this distance is defined as:

$$d(D_t, D_e) = \frac{1}{2m} \sum_{i=1}^{m} \min_{j=1}^{n} \|\mathbf{r}_t^i - \mathbf{r}_e^j\|_2^2. \quad (1)$$

Since $\mathbf{r}$ is obtained by concatenating independently normalized visual and textual features, the factor $1/2$ averages the squared discrepancy over the two modalities. With the distance $d(D_t, D_e)$ computed, we define a similarity-based discounting weight $W_{S_k}^t$ to mitigate the confounding effect of positive transfer. For the $k$-th evaluation dataset, the weight is inversely proportional to its semantic similarity to the current task:

$$S_k^t = \exp\left(-\beta \cdot d(D_t, D_e^{(k)})\right), \quad W_{S_k}^t = 1 - S_k^t. \quad (2)$$

Here, $\beta$ controls the discounting sharpness. The exponential form is inspired by the distance-to-similarity kernel (Van der Maaten & Hinton, 2008), providing a bounded monotonic calibration from semantic distance to transfer likelihood. A high $S_k^t$ results in a low $W_{S_k}^t$, discounting the dataset where positive transfer is likely dominant. Conversely, a high $W_{S_k}^t$ is assigned to dissimilar domains, treating performance drop there as an indicator of foundational damage.

These weights are applied to the transfer score $\Delta \mathcal{A}_k^t$, defined as the accuracy deviation from the original pre-trained baseline ($\mathcal{A}_k^0$) after learning task $t$:

$$\Delta \mathcal{A}_k^t = \mathcal{A}_k^t - \mathcal{A}_k^0. \quad (3)$$

For each incremental task $t$, we first compute $\text{SCR}_t$ as a weighted arithmetic mean over all zero-shot evaluation datasets:

$$\text{SCR}_t = \frac{\sum_{k=1}^{M} W_{S_k}^t \cdot \Delta \mathcal{A}_k^t}{\sum_{k=1}^{M} W_{S_k}^t}. \quad (4)$$

The final SCR score is obtained by averaging $\text{SCR}_t$ over all $T$ incremental tasks:

$$\text{SCR} = \frac{1}{T} \sum_{t=1}^{T} \text{SCR}_t. \quad (5)$$

A higher SCR score (closer to zero or positive) indicates that the model has maintained its foundational stability and zero-shot capability even on concepts strictly disjoint from the learned tasks.

## 4. Method

To address the dual-forgetting challenge, we propose *DFA-MoE*, a functionally heterogeneous PEFT framework. Unlike conventional MoE approaches that assign experts based on data partitions yet optimize them via a homogeneous loss, we architecturally decouple the functional roles. We design two functionally distinct pathways: a dedicated *Alignment Expert* is constrained via momentum-based contrastive learning to preserve the VLM's foundational feature space (targeting PKF), while *Plasticity Experts* are optimized via a composite objective. Specifically, the plasticity branch combines cross-entropy for new class discrimination with an auxiliary contrastive loss that leverages the momentum queue to recall historical task distributions (targeting IKF). This design effectively resolves the conflict between stability and plasticity by ensuring that foundational alignment and incremental history are preserved through dedicated mechanisms.

### 4.1. Preliminary: Mixture-of-Experts for Continual Learning

Recent adaptations of MoE for continual learning integrate parameter-efficient experts (*e.g.*, LoRA (Hu et al., 2022)) into frozen backbones (Yu et al., 2024a; 2025a). A routing network $r(\cdot)$ dynamically aggregates these experts $\{E_i\}_{i=1}^{N_E}$ via weighted summation:

$$\tilde{x} = \sum_{i=1}^{N_E} [r(x)]_i \cdot E_i(x). \quad (6)$$

Despite their plasticity, most approaches (Yu et al., 2024a; Zhou et al., 2025) treat experts homogeneously as task learners, neglecting preservation of pre-trained knowledge. While recent efforts (Jung & Kim, 2024) have begun to explore depth-based heterogeneity, they differ fundamentally from our proposed architecture, which implements parallel functional decoupling to balance alignment and plasticity.

### 4.2. DFA-MoE: Functionally Heterogeneous MoE for Dual Forgetting

To overcome the geometric conflict inherent in homogeneous MoEs, DFA-MoE introduces a functional decoupling. As illustrated in Figure 4 (Right), we structurally separate the architecture into two specialized pathways, each governed by a distinct objective. The *Alignment Pathway* employs a task-agnostic expert ($E_a$) to explicitly mitigate PKF. Driven by a *momentum-based contrastive loss*, it anchors new inputs to the pre-trained semantic space to maintain foundational stability. Conversely, the *Plasticity Pathway* utilizes task-specific experts ($E_s$) to address IKF and adaptation. This pathway adopts a dual strategy, optimizing cross-entropy for new class discrimination while leveraging an auxiliary contrastive loss—guided by the momentum

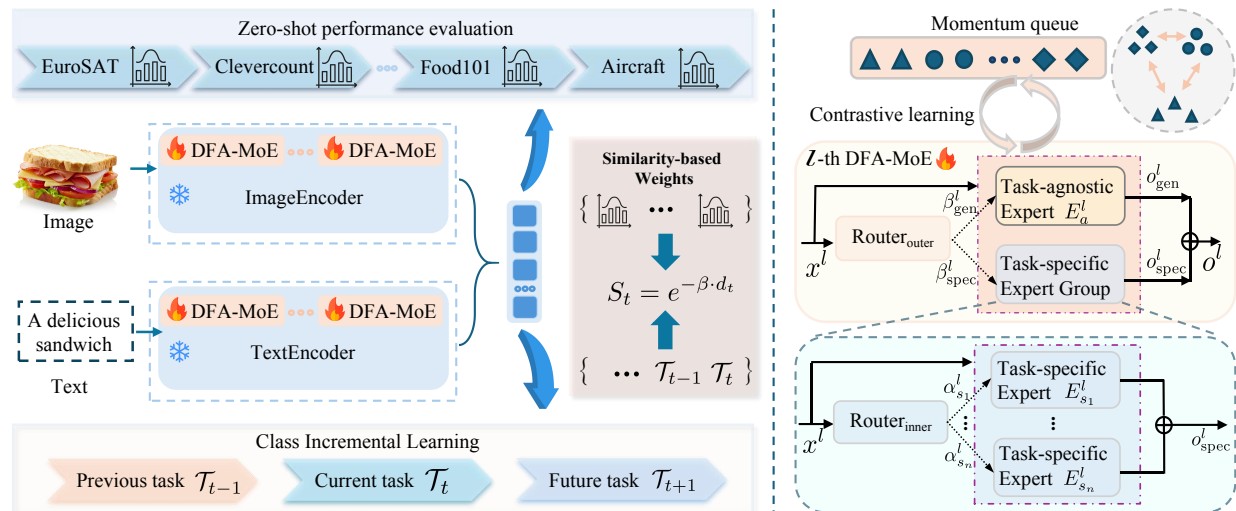

*Figure 4.* An overview of our proposed framework. Left: A fine-grained view of the DFA-CIL framework, depicting its parallel execution of CIL and zero-shot evaluations. The zero-shot performance is weighted by the task similarity score ($S_t$) for a more robust analysis. Right: The DFA-MoE method features two expert branches to address dual forgetting. A contrastive-trained Task-agnostic Expert preserves general knowledge, while a Task-specific Expert Group learns new concepts and retains historical context via composite optimization. An outer router dynamically balances their contributions.

queue—to preserve historical knowledge.

**The Alignment Pathway: Stabilizing via Momentum Contrast.** The primary goal of this pathway is to mitigate PKF by countering the feature drift caused by task-specific classification losses. To achieve this, the task-agnostic expert ($E_a$) is shielded from cross-entropy optimization and is instead trained exclusively via a contrastive objective. However, a critical challenge in applying contrastive learning to continual settings is the scarcity of diverse negative samples; without the vast pre-training corpus, the model risks representation collapse. To bridge this gap, we integrate a momentum queue that serves as a dynamic proxy for the global semantic space. Operated on a First-In-First-Out basis (see Supplementary Material B for details), this queue stores a rolling history of features, allowing $E_a$ to contrast current inputs against a diverse set of historical representations, thereby stabilizing the feature space against the narrow distribution of the current task. Crucially, this queue persists beyond the alignment stage, serving as a historical memory bank that the Plasticity Pathway leverages to mitigate IKF in the subsequent stage.

We employ a symmetric InfoNCE loss (Oord et al., 2018), where the negative set is constructed from both current mini-batch non-matches and the historical momentum queue. For an image anchor $v_i$ and its text pair $t_i$, the loss is defined as:

$$\mathcal{L}_{v \to t}^{(i)} = -\log \frac{\exp(\text{sim}(v_i, t_i)/\tau)}{\exp(\text{sim}(v_i, t_i)/\tau) + \sum_{k=1}^{K} \exp(\text{sim}(v_i, t_k^-)/\tau)}, \quad (7)$$

where $t_k^-$ represents negative samples from the momentum queue and the mini-batch, $\text{sim}(\cdot, \cdot)$ denotes cosine similar-

ity, and $\tau$ is the temperature. This objective compels $E_a$ to maintain robust image-text alignment by pulling corresponding pairs closer while pushing away a temporally broad set of negatives. The final objective averages the bidirectional losses across all $N$ pairs in the batch:

$$\mathcal{L}_{\text{con}} = \frac{1}{2N} \sum_{i=1}^{N} \left( \mathcal{L}_{v \to t}^{(i)} + \mathcal{L}_{t \to v}^{(i)} \right). \quad (8)$$

By continuously optimizing this alignment-centric objective, the expert $E_a$ effectively helps preserve the VLM's core generalization capabilities.

**The Plasticity Pathway: Hierarchical Routing.** With the foundational alignment secured by the first stage, the second stage focuses on the plasticity pathway. To acquire new class knowledge without disrupting the stabilized feature space, we freeze the task-agnostic expert ($E_a$) and exclusively update the task-specific expert group ($E_s = \{E_{s_1}, \ldots, E_{s_n}\}$) alongside the routing modules. As illustrated in Figure 4 (Right), for an input feature $x^l$ at layer $l$, the architecture employs a hierarchical mechanism to integrate knowledge.

First, the *frozen alignment pathway* produces a stable representation: $o_{\text{gen}}^l = E_a^l(x^l)$. Simultaneously, the *adaptive plasticity pathway* activates the trainable experts. An inner router ($r_{\text{inner}}^l$) computes a softmax-normalized gating vector ($\alpha_{s_1}^l, \ldots, \alpha_{s_n}^l$) to synthesize a task-specific output:

$$o_{\text{spec}}^l = \sum_{j=1}^{n} \alpha_{s_j}^l E_{s_j}^l(x^l). \quad (9)$$

Subsequently, an outer router ($r_{\text{outer}}^l$) acts as a global gate-keeper. It determines the optimal fusion weights ($\beta_{\text{gen}}^l, \beta_{\text{spec}}^l$)

*Table 1.* Comparison of all methods on the EuroSAT, OxfordFlower, and FER2013 datasets within our DFA-CIL framework. **Bold** indicates the best performance, and underline indicates the second best. An asterisk (*) marks methods we adapted for the CIL setting. For BWT and SCR, a higher value is better, indicating less forgetting.

| Datasets | Method | Venue | CIL | | Zero-shot Performance | | | | | | | |
| | | | | | Low-Similarity | | Mid-Similarity | | High-Similarity | | Overall | |
| | | | Last. (↑) | BWT (↑) | Last. (↑) | SCR (↑) | Last. (↑) | SCR (↑) | Last. (↑) | SCR (↑) | Last. (↑) | SCR (↑) |
|---|---|---|---|---|---|---|---|---|---|---|---|---|
| EuroSAT | CLIP Zero-shot (Radford et al., 2021) | ICML'21 | – | – | 61.91 | 0.00 | 58.87 | 0.00 | 62.10 | 0.00 | 60.96 | 0.00 |
| | LwF (Li & Hoiem, 2017) | TPAMI'17 | 58.93 | -21.36 | 56.03 | -4.34 | 52.20 | -4.46 | 53.53 | -7.29 | 53.92 | -5.18 |
| | iCaRL (Rebuffi et al., 2017) | CVPR'17 | 54.03 | -20.46 | 54.21 | -5.51 | 48.85 | -8.22 | 50.48 | -10.82 | 51.18 | -7.85 |
| | LwF-VR (Ding et al., 2022) | ArXiv'22 | 55.45 | -33.76 | 54.89 | -5.72 | 52.83 | -4.74 | 52.91 | -7.00 | 53.54 | -5.75 |
| | MoE4Adapter (Yu et al., 2024a) | CVPR'24 | 62.52 | -15.56 | 56.99 | -4.92 | 58.09 | -4.43 | 61.50 | -3.30 | 58.86 | -4.16 |
| | Chordprompt* (Wang & Chen, 2025) | ECML'25 | 62.72 | -21.31 | 49.14 | -11.70 | 41.03 | -10.36 | 44.31 | -13.37 | 44.82 | -11.72 |
| | DMNSP (Kang et al., 2025) | ICCV'25 | 69.64 | -3.40 | 56.81 | -3.93 | 57.76 | -2.26 | 61.89 | -1.27 | 58.82 | -2.65 |
| | GIFT (Wu et al., 2025) | CVPR'25 | 69.52 | **-2.57** | 58.79 | 1.53 | 57.81 | -0.95 | 61.33 | -0.61 | 59.31 | -1.04 |
| | DFA-MoE (Ours) | - | **72.25** | -2.96 | **59.62** | **-1.41** | **59.06** | **-0.44** | **61.95** | **-0.39** | **60.21** | **-0.79** |
| OxfordFlower | CLIP Zero-shot (Radford et al., 2021) | ICML'21 | – | – | 49.94 | 0.00 | 60.88 | 0.00 | 69.16 | 0.00 | 59.99 | 0.00 |
| | LwF (Li & Hoiem, 2017) | TPAMI'17 | 67.50 | -13.47 | 48.49 | -1.25 | 59.65 | -0.77 | 69.06 | 0.31 | 59.06 | -0.78 |
| | iCaRL (Rebuffi et al., 2017) | CVPR'17 | 68.77 | -12.63 | 47.95 | -1.38 | 59.08 | -1.10 | 69.02 | 0.27 | 58.68 | -0.86 |
| | LwF-VR (Ding et al., 2022) | ArXiv'22 | 65.71 | -14.01 | 48.55 | -1.11 | 59.35 | -0.93 | 69.05 | 0.21 | 58.98 | -0.74 |
| | MoE4Adapter (Yu et al., 2024a) | CVPR'24 | 67.72 | **-5.39** | 48.52 | -1.12 | 60.38 | -0.68 | 69.10 | -1.00 | 59.33 | -0.75 |
| | Chordprompt* (Wang & Chen, 2025) | ECML'25 | 31.18 | -43.28 | 33.60 | -18.48 | 38.25 | -22.30 | 47.06 | -24.42 | 39.63 | -21.28 |
| | DMNSP (Kang et al., 2025) | ICCV'25 | 66.26 | -12.71 | 46.59 | -2.48 | 59.58 | -0.75 | 69.20 | **0.34** | 58.46 | -1.03 |
| | GIFT (Wu et al., 2025) | CVPR'25 | 70.21 | -10.28 | 48.62 | -1.08 | 56.61 | -2.88 | 69.01 | 0.31 | 58.42 | -0.86 |
| | DFA-MoE (Ours) | - | 70.40 | -6.37 | 48.69 | -1.02 | **60.44** | **-0.65** | **69.25** | -0.01 | **59.46** | **-0.66** |
| FER2013 | CLIP Zero-shot (Radford et al., 2021) | ICML'21 | – | – | 51.96 | 0.00 | 66.55 | 0.00 | 64.44 | 0.00 | 60.99 | 0.00 |
| | LwF (Li & Hoiem, 2017) | TPAMI'17 | 47.00 | -41.12 | 42.31 | -4.81 | 49.75 | -10.30 | 45.71 | -8.13 | 45.93 | -7.48 |
| | iCaRL (Rebuffi et al., 2017) | CVPR'17 | 45.49 | -34.56 | 42.86 | -4.92 | 52.70 | -9.08 | 52.86 | -5.82 | 49.48 | -6.54 |
| | LwF-VR (Ding et al., 2022) | ArXiv'22 | 11.74 | -52.33 | 40.12 | -5.90 | 41.24 | -13.40 | 36.96 | -11.43 | 39.45 | -9.79 |
| | MoE4Adapter (Yu et al., 2024a) | CVPR'24 | 60.59 | -14.88 | 47.88 | -3.16 | 64.90 | -1.54 | 63.06 | **-0.70** | 58.61 | -2.00 |
| | Chordprompt* (Wang & Chen, 2025) | ECML'25 | 43.49 | -22.27 | 36.26 | -16.57 | 50.45 | -15.16 | 55.44 | -8.88 | 47.28 | -14.20 |
| | DMNSP (Kang et al., 2025) | ICCV'25 | 56.35 | -27.95 | 48.33 | -3.15 | 64.72 | -1.20 | 62.73 | -1.67 | 58.59 | -2.23 |
| | GIFT (Wu et al., 2025) | CVPR'25 | 44.98 | -16.21 | 48.71 | -3.16 | 65.65 | -1.08 | 63.45 | -0.98 | 59.27 | -1.83 |
| | DFA-MoE (Ours) | - | **62.37** | **-13.21** | **48.89** | -2.84 | **65.69** | **-1.06** | **63.52** | -0.80 | **59.38** | **-1.74** |

to balance the contribution of stable retention versus incremental plasticity. The final output $o^l$ is the weighted sum:

$$o^l = \beta^l_{\text{gen}} o^l_{\text{gen}} + \beta^l_{\text{spec}} o^l_{\text{spec}}, \quad (10)$$

which is added as a residual to the block's main stream. During this stage, all trainable parameters (experts in $E_s$ and both routers) are optimized via a composite objective. Let $\mathcal{L}_{\text{CE}}$ denote the standard cross-entropy loss for the current task. The final objective combines this discrimination loss with the auxiliary contrastive term:

$$\mathcal{L}_{\text{total}} = \mathcal{L}_{\text{CE}} + \lambda \mathcal{L}_{\text{con}}, \quad (11)$$

where $\mathcal{L}_{\text{con}}$ follows the identical calculation process as the Alignment Pathway (Eq. 8).

At inference, DFA-MoE operates via a unified forward pass. The hierarchical routing automatically modulates the reliance on foundational versus specific experts for each instance, eliminating the need for external task identifiers or manual mode switching.

## 5. Experiments

### 5.1. Experimental Setup

**Datasets.** Our experiments are conducted according to our DFA-CIL framework, which is built upon a diverse pool of **25 datasets**. Among the 25 datasets, 22 datasets contain sufficient classes to serve as CIL sources, while three

binary datasets are used only as zero-shot evaluation targets. This collection is primarily drawn from the original CLIP evaluation sets (Radford et al., 2021) to help maintain data that is effectively unseen for evaluation, and it spans a wide spectrum of visual domains. Our experimental design is twofold. For an in-depth, granular analysis, we select three representative datasets to serve as the CIL source tasks: OxfordFlower (Nilsback & Zisserman, 2008), EuroSAT (Helber et al., 2019), and FER2013 (Goodfellow et al., 2013). These are chosen to represent distinct types of specialized knowledge a VLM might need to acquire: fine-grained classification on natural RGB images (OxfordFlower), adaptation to a non-natural image domain (EuroSAT), and robustness on non-RGB images (FER2013). Based on their class counts, these are partitioned into 17, 5, and 4 incremental tasks.

Moreover, to better understand how PKF manifests on domains with lower similarity (where the erosion of foundational knowledge is more pronounced), we partition the remaining 24 evaluation datasets into Low-, Mid-, and High-Similarity bins based on their similarity with the CIL source task. The top eight datasets form the High-Similarity group, the middle eight form the Mid-Similarity group, and the bottom eight form the Low-Similarity group. In addition, to provide a holistic evaluation, we follow the aforementioned "Leave-One-Out" procedure, which is also conducted in this experiment. More details on the datasets are provided in the Supplementary Material A.

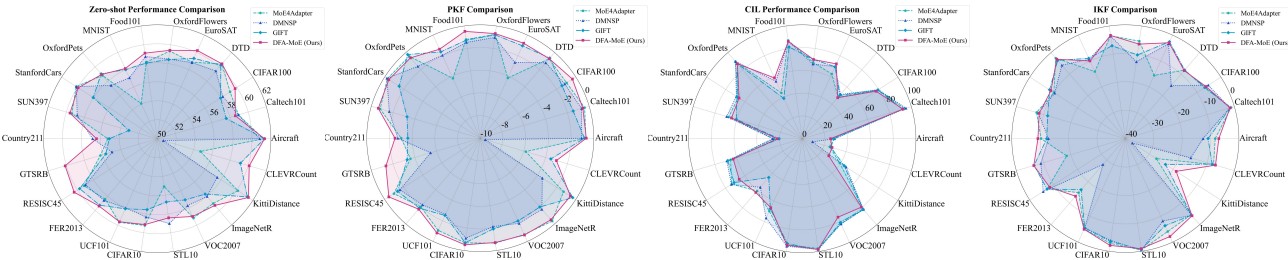

*Figure 5.* Radar charts comparing the final performance of DFA-MoE against baselines **across all 22 CIL datasets** in DFA-CIL framework. A larger polygon area indicates better performance. The charts clearly show DFA-MoE's overall superior ability to preserve pre-trained knowledge (Zero-shot Last Acc. and SCR) while maintaining competitive CIL performance.

**Implementation Details.** DFA-MoE is built upon the pre-trained ViT-B/16 CLIP model, with its backbone parameters kept frozen throughout all experiments. The experts in both the task-agnostic ($E_a$) and task-specific ($E_s$) pathways are implemented as lightweight MLP adapters. Following MoE4Adapter (Yu et al., 2024a), the task-specific expert group ($E_s$) is configured with $n = 2$ experts, and the momentum queue length is set to 128. For the composite optimization in the plasticity pathway, the balancing coefficient $\lambda$ for the contrastive loss is set to 0.001. $\beta$ is fixed to 1.0 in all experiments. Further details are provided in the Supplementary Material B.

## 5.2. Comparison Results

We compare DFA-MoE against a comprehensive suite of baselines on three diverse CIL datasets in Table 1. The baselines include both full fine-tuning methods (LwF (Li & Hoiem, 2017), iCaRL (Rebuffi et al., 2017), LwF-VR (Ding et al., 2022), GIFT (Wu et al., 2025)), which update the entire CLIP; and various PEFT-based approaches (Chord-prompt (Wang & Chen, 2025), MoE4Adapter (Yu et al., 2024a) and DMNSP (Kang et al., 2025)). To create a fair comparison and highlight the challenges of task-agnostic learning, we adapt Chordprompt (Wang & Chen, 2025)—a leading method from the task-id-dependent MTIL setting—to our more demanding task-id-free protocol (denoted as Chordprompt*).

The results underscore DFA-MoE's superior balance in addressing the dual-forgetting challenge. Regarding IKF, DFA-MoE consistently sets a new state-of-the-art by achieving the highest final CIL accuracy across all three datasets: 72.25% on EuroSAT, 70.40% on OxfordFlower, and 62.37% on FER2013 (see "CIL-Last." column in Table 1). Crucially, DFA-MoE demonstrates a unique capability in mitigating PKF, breaking the fundamental stability-efficiency trade-off that plagues existing methods. While replay-based methods like GIFT preserve PKF well, they incur massive parameter and computational overhead (see Table 2). Conversely, other efficient PEFT methods like MoE4Adapter and DMNSP suffer from severe foundational erosion on domain-shifted

tasks. DFA-MoE is the only method that maintains near-perfect foundational knowledge, tracking the original CLIP baseline closely while remaining highly parameter-efficient.

This superiority is further validated by our granular analysis using the SCR metric. The Low- and Mid-Similarity groups serve as the most rigorous testbeds for genuine knowledge retention, as their performance is least influenced by positive transfer from new tasks. In these critical groups, DFA-MoE achieves the highest accuracy and the best SCR scores. These results confirm that DFA-MoE's effectiveness is not an artifact of local generalization on similar domains, but a robust capability to preserve the VLM's foundational zero-shot universality.

**Holistic Performance Analysis Across DFA-CIL.** To evaluate model robustness beyond individual domains, we conduct an extensive cross-task analysis using a "leave-one-out" protocol. We treat each of the 22 datasets in our pool as a CIL source task in turn, evaluating the resulting PKF on the remaining 24 datasets. As visualized in the radar charts (Figure 5), DFA-MoE exhibits clear dominance in preserving pre-trained knowledge. Its performance polygons for both Zero-shot Accuracy and SCR consistently encompass those of all baselines, demonstrating superior PKF mitigation that is agnostic to the training domain. Regarding IKF, while overall accuracy is competitive among top-tier methods, DFA-MoE's significantly larger polygon in the BWT chart signifies a robust ability to protect previously acquired tasks. Collectively, these results confirm that DFA-MoE achieves the most effective balance in addressing dual forgetting across a vast spectrum of visual domains.

**Forgetting Trajectory Analysis.** To further validate DFA-MoE's preservation of pre-trained knowledge, we conduct an analysis of its performance trajectory as new knowledge is acquired. As shown in Figure 6, DFA-MoE's performance curve remains consistently high and remarkably flat, demonstrating that knowledge preservation is a continuous property, not merely a final-state outcome. In contrast, even strong baselines like MoE4Adapter, DMNSP, and GIFT

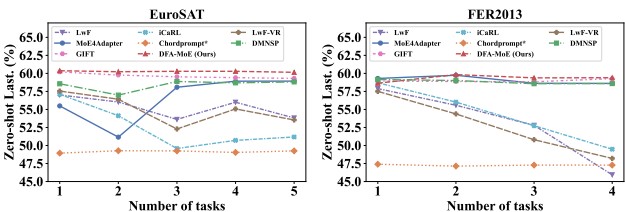

*Figure 6.* Trajectory of Zero-Shot Last Accuracy on the EuroSAT and FER2013 CIL tasks. The plot tracks the performance of all methods after each incremental task. More trajectory analyses on additional datasets are provided in the Supplementary Material F.

*Table 2.* Computational cost comparison of trainable parameters, training time per iteration, and foundational stability. Avg. SCR denotes the mean Similarity-Calibrated Retention score across the 22 distinct experiments visualized in our radar chart analysis.

| Method | Params. (↓) | Time (↓) | Avg. SCR (↑) |
|---|---|---|---|
| MoE4Adapter (Yu et al., 2024a) | 4.03M | **0.16 s/it** | -1.57 |
| DMNSP (Kang et al., 2025) | 1.98M | 0.28 s/it | -2.23 |
| GIFT (Wu et al., 2025) | 149.62M | 0.88 s/it | -1.69 |
| DFA-MoE (Ours) | **1.05M** | 0.25 s/it | **-0.72** |

*Table 3.* Ablation study demonstrating the incremental benefits of each core component. Comparing from a homogeneous MoE baseline, the sequential addition of the heterogeneous architecture, the Contrastive Learning objective (+CL), and the Momentum Queue (+Q) yields consistent performance gains.

| Variant | CIL | | Zero-shot Performance | | | | | | |
|---|---|---|---|---|---|---|---|---|---|
| | | | Low-Similarity | | Mid-Similarity | | High-Similarity | | Overall |
| | Last. | BWT | Last. | SCR | Last. | SCR | Last. | SCR | Last. SCR |
| Homogeneous MoE | 62.52 | -15.56 | 56.99 | -4.92 | 58.09 | -4.43 | 61.50 | -3.30 | 58.86 -4.16 |
| Heterogeneous MoE | 69.58 | -3.30 | 59.47 | -1.78 | 58.31 | -0.56 | 61.72 | -0.84 | 59.70 -1.11 |
| + CL | 71.74 | -3.18 | 59.58 | -1.58 | 58.72 | -0.52 | 61.84 | -0.51 | 60.04 -0.85 |
| + CL + Q (Ours) | **72.25** | **-2.96** | **59.62** | **-1.41** | **59.06** | **-0.44** | **61.95** | **-0.39** | **60.21 -0.79** |

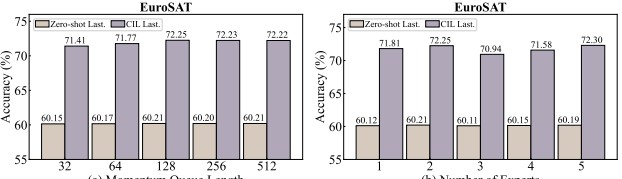

*Figure 7.* Ablation study on EuroSAT CIL. Zero-shot and CIL "Last." accuracies are plotted against two hyperparameters: (a) momentum queue length and (b) number of task-specific experts.

exhibit significant performance variation and a downward trend as more tasks are learned. This suggests their preservation mechanisms are fragile and degrade under continued exposure to domain-specific data. DFA-MoE's stability stems directly from its core design, which explicitly decouples the learning of new concepts from the preservation of foundational knowledge, thereby avoiding this trade-off.

**Computational Cost Analysis.** As shown in Table 2, DFA-MoE achieves superior parameter efficiency, requiring only 1.05M parameters—nearly half that of DMNSP (1.98M). Regarding computational speed, our method (0.25 s/it) proves highly efficient, outperforming DMNSP (0.28 s/it) and operating over $3.5\times$ faster than the replay-based GIFT (0.88 s/it). While MoE4Adapter (0.16 s/it) achieves the fastest raw training speed, this slight advantage comes at the heavy cost of foundational instability (a relatively lowe SCR of -1.57 among PEFT methods). In contrast, DFA-MoE maintains a comparable lightweight training speed while achieving the highest stability (-0.72). A detailed computational cost analysis is provided in Supplementary Material D. These results confirm that DFA-MoE provides the most effective balance between training efficiency and foundational preservation for sustainable VLM continual learning.

### 5.3. Ablation Study

**Impact of Heterogeneous Design and Training Strategy.** Our progressive ablation study (Table 3) validates the contribution of each design component. First, replacing the standard Homogeneous MoE with our Heterogeneous MoE design yields a dramatic performance gain. This initial

step improves CIL accuracy by over 7% and, more critically, boosts the Overall SCR score from a severe -4.16% to -1.11%, confirming that architectural separation is fundamental. Next, applying Contrastive Learning to the task-agnostic expert specifically targets PKF, further improving the Overall SCR to -0.85%. Finally, the Momentum Queue unlocks a pivotal dual benefit: it stabilizes alignment via diverse negatives and enables the auxiliary contrastive loss for plasticity experts to recall historical distributions. This mechanism effectively mitigates IKF, pushing CIL accuracy to a peak of 72.25% while achieving the best SCR of -0.79%. This step-by-step improvement demonstrates that each component is essential for our final state-of-the-art performance. We further visualize the learned inner- and outer-router weight distributions in Supplementary Material C, showing adaptive and non-collapsed routing behavior across transformer layers.

**Impact of Momentum Queue Length.** We analyze the impact of the momentum queue's capacity, shown in Figure 7 (a). A critical advantage of our design is that the queue length is fixed, ensuring a constant and low memory overhead that does not scale with incremental tasks. Despite this minimal storage cost, the performance gains are substantial: a compact length of 128 boosts CIL and Zero-shot accuracy to peaks of 72.25% and 60.21%, respectively. This demonstrates the high effectiveness of our approach, where a concise, static historical pool provides sufficient negative diversity to significantly mitigate forgetting. Furthermore, extending the length to 256 or 512 yields negligible gains, confirming that our method achieves robust alignment saturation without requiring memory-intensive buffers.

**Impact of the Number of Task-Specific Experts.** We investigate the effect of the number of experts, $n$, by varying its value from 1 to 5, with results presented in Figure 7 (b). The two-expert configuration reaches a peak Zero-shot Last accuracy of 60.21% and a CIL Last accuracy of 72.25%. Although using five experts yields a small improvement in CIL accuracy to 72.3%, it results in slightly lower zero-shot performance and incurs substantial parameter and computational overhead. The two-expert group offers a superior trade-off between performance, parameter efficiency, and computational cost. We therefore adopt $n = 2$ as our default configuration, consistent with prior work such as MoE4Adapter (Yu et al., 2024a).

## 6. Conclusion

This paper investigates dual forgetting in Vision-Language Continual Learning, where models suffer from both Incremental Knowledge Forgetting (IKF) and Pre-trained Knowledge Forgetting (PKF). To enable a more reliable evaluation, we design the DFA-CIL framework to measure IKF and PKF in a practical setting, featuring the Similarity-Calibrated Retention metric. With this framework, we propose DFA-MoE, a PEFT-based method that decouples new knowledge acquisition from pre-trained knowledge preservation to achieve a state-of-the-art balance in mitigating IKF and PKF. Extensive experiments validate the effectiveness of DFA-MoE. In future work, we will apply DFA-CIL to investigate dual forgetting across a broader spectrum of VLMs.

## Impact Statement

This paper presents work whose goal is to advance the field of machine learning. There are many potential societal consequences of our work, none of which we feel must be specifically highlighted here.

## Acknowledgements

This work is supported in part by the National Natural Science Foundation of China (62576160, 62192783), Fundamental Research Funds for the Central Universities (KG202514), 111 Center (B26023), and the Australian Research Council's Discovery Project (DP220101784), Fundamental and Interdisciplinary Disciplines Breakthrough Plan of the Ministry of Education of China (JYB2025XDXM118).

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

# Supplementary Material: Don't Forget Why You Started: Tackling Dual Forgetting in Vision-Language Continual Learning

## A. Dataset and Task Partitioning Details

**Overview and Categorization of Datasets**    The proposed DFA-CIL evaluation framework is built upon a comprehensive and diverse pool of 25 datasets. This collection is primarily drawn from the original CLIP (Radford et al., 2021) evaluation sets to ensure that the data is genuinely "unseen" by the pre-trained vision-language model (VLM), thereby providing a fair and rigorous testbed. As summarized in Table 4, the pool spans a wide spectrum of visual domains, which can be broadly categorized into five groups:

- **General Object Recognition:** This category comprises foundational benchmarks for computer vision, used to evaluate a model's ability to recognize common objects. It includes datasets such as CIFAR10/100 (Krizhevsky et al., 2009), VOC2007 (Everingham et al.), Caltech101 (Fei-Fei et al., 2006), STL10 (Coates et al., 2011), and the artistically-rendered ImageNet-R (Hendrycks et al., 2021).

- **Fine-grained Visual Classification:** This group of datasets is designed to challenge a model's ability to distinguish between subtle differences among subclasses. This category includes datasets for food classification (Food101 (Bossard et al., 2014)), car models (StanfordCars (Krause et al., 2013)), aircraft models (Aircraft (Maji et al., 2013)), pet breeds (OxfordPets (Parkhi et al., 2012)), and flower species (OxfordFlower (Nilsback & Zisserman, 2008)).

- **Scene and Environmental Recognition:** The focus of tasks in this category is on understanding the overall environment or geospatial context of an image. It includes the large-scale scene classification dataset SUN397 (Xiao et al., 2010), satellite remote sensing imagery (EuroSAT (Helber et al., 2019) and RESISC45 (Cheng et al., 2017)), autonomous driving scenes (KITTI (Geiger et al., 2013)), and geographically-associated images (Country211 (Radford et al., 2021)).

- **Domain-Specific Recognition:** This category covers datasets from specialized fields or those with unique image characteristics, often requiring domain-specific knowledge. It includes datasets for texture classification (DTD (Cimpoi et al., 2014)), handwritten digit recognition (MNIST (LeCun et al., 2010)), facial expression recognition (FER2013 (Goodfellow et al., 2013)), traffic sign recognition (GTSRB (Stallkamp et al., 2011)), medical histopathological imaging (PCam (Veeling et al., 2018)), and human action recognition (UCF101 (Soomro et al., 2012)).

- **Visual Reasoning and Multimodal Classification:** This set of tasks extends beyond simple image classification, requiring higher-level capabilities such as reasoning or understanding combined image-text content. This category includes synthetic imagery for visual counting (CLEVRcounts (Johnson et al., 2017)), multimodal hate speech detection (Hatefulmemes (Kiela et al., 2020)), and sentiment analysis from text (SST2 (Socher et al., 2013)).

This extensive diversity allows for a holistic assessment of how dual forgetting manifests across varying degrees of domain shift and task complexity.

**CIL Task Partitioning Protocol**    Our evaluation operates under a strict "Leave-One-Out" protocol: in each experiment, one dataset is designated as the Class-Incremental Learning (CIL) source task, while the remaining 24 datasets constitute the zero-shot evaluation pool ($D_{pool}$) to assess Pre-trained Knowledge Forgetting (PKF). When a dataset serves as the CIL source, it is partitioned into a sequence of distinct tasks. As detailed in the CIL Task Partitioning column of Table 4 in this supplementary material A, we use the notation *IncX-TaskY*, where *Y* is the total number of tasks and *X* is the standard number of new classes per task.

To handle datasets where the total number of classes is not perfectly divisible by the number of tasks (marked with a dagger $^\dagger$ in the table), we employ a "front-loading" strategy where the remainder classes are distributed among the initial tasks. Specifically:

- Food101 (101 classes, 10 tasks): Partitioned into 1 task of 11 classes followed by 9 tasks of 10 classes.

- SUN397 (397 classes, 18 tasks): Partitioned into 1 task of 23 classes followed by 17 tasks of 22 classes.

*Table 4.* Details of the 25 datasets used in the DFA-CIL framework. For each dataset serving as a CIL source, the partitioning scheme is detailed in the **CIL Task Partitioning** column, following the format *IncX-TaskY*. Here, *Y* denotes the total number of incremental tasks, and *X* represents the standard number of new classes introduced in each task. A dagger ($^\dagger$) indicates datasets where the total number of classes is not perfectly divisible by the number of tasks. In these cases, the extra classes are distributed among the initial tasks.

| Dataset | Classes | Train size | Test size | Domain/Task Type | Image Characteristics Description | CIL Task Partitioning |
|---|---|---|---|---|---|---|
| Food101 (Bossard et al., 2014) | 101 | 50,500 | 30,300 | Fine-grained Visual Classification | High-resolution natural images of 101 food categories. | Inc10-Task10$^\dagger$ |
| CIFAR10 (Krizhevsky et al., 2009) | 10 | 40,000 | 10,000 | General Object Recognition | Low-resolution (32x32) natural images of 10 classes. | Inc2-Task5 |
| CIFAR100 (Krizhevsky et al., 2009) | 100 | 40,000 | 10,000 | General Object Recognition | Low-resolution (32x32) natural images of 100 classes. | Inc10-Task10 |
| SUN397 (Xiao et al., 2010) | 397 | 15,880 | 19,850 | Scene and Environmental Recognition | Images from 397 different scene categories (*e.g.*, airport, forest). | Inc22-Task18$^\dagger$ |
| StanfordCars (Krause et al., 2013) | 196 | 6,509 | 8,041 | Fine-grained Visual Classification | Images of 196 classes of cars (make, model, year). | Inc14-Task14 |
| Aircraft (Maji et al., 2013) | 100 | 3,334 | 3,333 | Fine-grained Visual Classification | Images of 100 different aircraft models. | Inc10-Task10 |
| VOC2007 (Everingham et al.) | 20 | 5,011 | 4,952 | General Object Recognition | Natural images of 20 common object categories. | Inc4-Task5 |
| DTD (Cimpoi et al., 2014) | 47 | 2,820 | 1,692 | Domain-Specific Recognition | Describable Textures in the Wild, containing various texture patterns. | Inc9-Task5$^\dagger$ |
| OxfordPets (Parkhi et al., 2012) | 37 | 2,944 | 3,669 | Fine-grained Visual Classification | Images of 37 different breeds of cats and dogs. | Inc6-Task6$^\dagger$ |
| Caltech101 (Fei-Fei et al., 2006) | 101 | 4,128 | 2,465 | General Object Recognition | Images of 101 object categories. | Inc10-Task10$^\dagger$ |
| OxfordFlower (Nilsback & Zisserman, 2008) | 102 | 4,093 | 2,463 | Fine-grained Visual Classification | Images of 102 common flower species. | Inc6-Task17 |
| MNIST (LeCun et al., 2010) | 10 | 48,000 | 10,000 | Domain-Specific Recognition | Grayscale images of handwritten digits from 0 to 9. | Inc2-Task5 |
| FER2013 (Goodfellow et al., 2013) | 7 | 22,968 | 7,178 | Domain-Specific Recognition | Grayscale facial images depicting 7 basic emotions. | Inc1-Task4$^\dagger$ |
| STL10 (Coates et al., 2011) | 10 | 3,200 | 1,000 | General Object Recognition | Higher-resolution (96x96) images of 10 classes. | Inc2-Task5 |
| EuroSAT (Helber et al., 2019) | 10 | 13,500 | 8,100 | Scene and Environmental Recognition | Sentinel-2 satellite images of 10 land use/cover classes. | Inc2-Task5 |
| RESISC45 (Cheng et al., 2017) | 45 | 15,750 | 94,50 | Scene and Environmental Recognition | Remote sensing images of 45 scene classes (*e.g.*, airplane, lake). | Inc5-Task9 |
| GTSRB (Stallkamp et al., 2011) | 43 | 19,604 | 11,763 | Domain-Specific Recognition | Real-world images of German traffic signs. | Inc7-Task6$^\dagger$ |
| KITTI (Geiger et al., 2013) | 4 | 4,787 | 1,496 | Scene and Environmental Recognition | Street scene images taken from a car's perspective. | Inc2-Task2 |
| Country211 (Radford et al., 2021) | 211 | 31,650 | 21,100 | Scene and Environmental Recognition | Various scenes and objects associated with 211 countries. | Inc21-Task10$^\dagger$ |
| PCam (Veeling et al., 2018) | 2 | 209,767 | 32,768 | Domain-Specific Recognition | Histopathological scans of lymph node sections for tumor detection. | Not applicable (2 classes) |
| UCF101 (Soomro et al., 2012) | 101 | 7,639 | 3,783 | Domain-Specific Recognition | Image frames extracted from videos of 101 human actions. | Inc10-Task10$^\dagger$ |
| CLEVRcounts (Johnson et al., 2017) | 8 | 55,999 | 15,000 | Visual Reasoning and Multimodal | Synthetic images with multiple 3D objects for counting tasks. | Inc2-Task4 |
| Hatefulmemes (Kiela et al., 2020) | 2 | 8,500 | 500 | Visual Reasoning and Multimodal | Memes containing images and text for hate speech detection. | Not applicable (2 classes) |
| SST2 (Socher et al., 2013) | 2 | 6,920 | 1,821 | Visual Reasoning and Multimodal | Sentences from movie reviews for sentiment classification. | Not applicable (2 classes) |
| ImageNet-R (Hendrycks et al., 2021) | 200 | 19,197 | 5,998 | General Object Recognition | Renditions of 200 ImageNet classes in various artistic styles. | Inc20-Task10 |

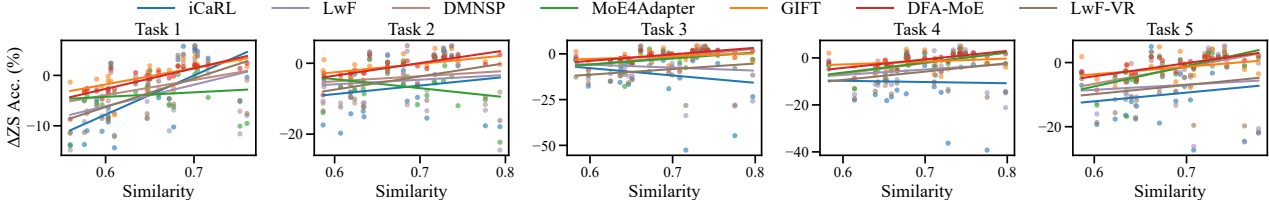

*Figure 8.* Visualization of similarity-induced bias across all incremental tasks on the **EuroSAT** dataset. The sub-figures report the change in zero-shot accuracy (ΔZS Acc.) on each evaluation domain after completing each of the 5 incremental tasks respectively. In each plot, there are 24 points, each representing a specific zero-shot evaluation domain. The x-axis denotes the domain's similarity to the current CIL task, while the y-axis tracks the change in zero-shot accuracy. The colored lines represent the linear regression fits for the specific methods listed in the legend at the top of the figure, highlighting the positive correlation between similarity and zero-shot degradation.

- DTD (47 classes, 5 tasks): Partitioned into 2 tasks of 10 classes followed by 3 tasks of 9 classes.

- OxfordPets (37 classes, 6 tasks): Partitioned into 1 task of 7 classes followed by 5 tasks of 6 classes.

- Caltech101 (101 classes, 10 tasks): Partitioned into 1 task of 11 classes followed by 9 tasks of 10 classes.

- FER2013 (7 classes, 4 tasks): Partitioned into 3 tasks of 2 classes followed by 1 task of 1 class.

- GTSRB (43 classes, 6 tasks): Partitioned into 1 task of 8 classes followed by 5 tasks of 7 classes.

- Country211 (211 classes, 10 tasks): Partitioned into 1 task of 22 classes followed by 9 tasks of 21 classes.

- UCF101 (101 classes, 10 tasks): Partitioned into 1 task of 11 classes followed by 9 tasks of 10 classes.

Finally, datasets with extremely few classes, specifically those with only 2 classes (PCam, HatefulMemes, SST2), are deemed unsuitable for meaningful incremental partitioning. Consequently, these datasets serve exclusively as zero-shot evaluation targets and are never used as CIL sources in our experimental setup. For a comprehensive breakdown of the specific partitioning schemes for all datasets, please refer to Table 4 in this supplementary material.

## B. Implementation Details

Our method utilizes the pre-trained ViT-B/16 CLIP model, with the backbone frozen. Structurally, both the task-agnostic expert ($E_a$) and the task-specific experts ($E_s$) are implemented as Bottleneck Adapters consisting of a down-projection,

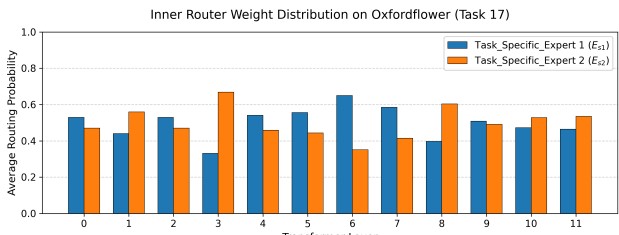 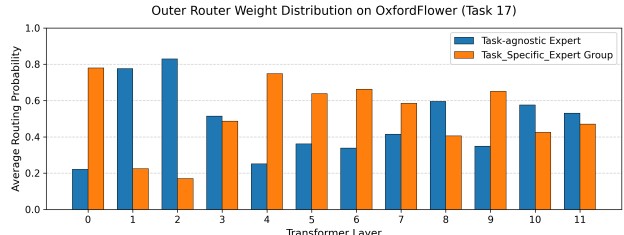

*Figure 9.* Layer-wise routing probabilities on OxfordFlower Task 17. Left: inner-router weights over the two task-specific experts. Right: outer-router weights over the task-agnostic expert and the task-specific expert group.

a non-linear activation, and an up-projection. The routing mechanisms are tailored to their distinct roles: the top-level outer router is designed as a Dense Softmax Grouped Gating Network, which learns continuous importance weights to balance the two pathways; conversely, the inner router within the task-specific group is implemented as a Noisy Top-$k$ Gating Network. For the task-specific group including two experts, we set $k = 2$, allowing the router to dynamically weight both experts while injecting noise during training to enhance robustness. The momentum queue is maintained using a strict First-In-First-Out (FIFO) policy to ensure a dynamic yet stable representation of historical distributions. Specifically, after the training of each incremental task is completed, we randomly sample 32 feature representations from the current task's data. These features are pushed into the queue, while the oldest 32 features are simultaneously removed (dequeued). This mechanism ensures the queue serves as a rolling memory bank without storing raw images, thus preserving the replay-free setting

All trainable parameters are optimized using the AdamW optimizer with a batch size of 128. The model is trained for a single epoch for each incremental task. Regarding hyperparameters, the task-agnostic expert ($E_a$) uses a bottleneck dimension of 16 and a constant learning rate of 1e-5, with the contrastive learning temperature ($\tau$) set to 0.15. The task-specific experts ($E_s$) use a bottleneck dimension of 8; their text-side learning rate is fixed at 1e-5, while the visual-side learning rate is searched from $\{5e\text{-}4, 1e\text{-}3\}$. Finally, the inner router is trained with a learning rate of 1e-3, while the outer router uses a learning rate of 1e-5. All experiments are conducted on NVIDIA GeForce RTX 3090 GPUs.

## C. Router Weight Distribution Analysis

To further illustrate the behavior of the routing modules, we visualize the layer-wise routing probabilities of the inner and outer routers on the final incremental task of OxfordFlower. Figure 9 reports the learned routing weights averaged over the evaluation set.

As shown in Figure 9 (left), the inner router assigns non-trivial weights to both task-specific experts across all transformer layers. The two experts remain relatively balanced, while their dominance varies across depth. This indicates that the inner router does not collapse to a single expert, but instead learns complementary expert allocation within the task-specific pathway. Figure 9 (right) shows that the outer router adaptively balances the task-agnostic expert and the task-specific expert group in a layer-wise manner. Some layers place greater emphasis on the task-agnostic pathway, while others favor the task-specific pathway, and several layers remain close to a balanced mixture. This indicates an adaptive layer-wise weight distribution between the task-agnostic pathway and the task-specific pathway.

Overall, these results show that the two routers learn distinct but complementary behaviors: the inner router coordinates collaboration within the task-specific branch, while the outer router adaptively fuses stable general knowledge and task-adaptive specialization.

## D. Computational Overhead Analysis

We further conduct a more detailed computational overhead analysis of DFA-MoE and competitive baselines on the EuroSAT CIL benchmark using the CLIP ViT-B/16 backbone. We report peak GPU memory, training time, and per-sample FLOPs. Besides the per-epoch cost, we also report the per-task cost, which measures the cumulative computation required to finish training on one incremental task under each method's original training schedule.

As shown in Table 5, DFA-MoE introduces higher per-epoch overhead than MoE4Adapter due to the contrastive alignment

*Table 5.* Computational overhead comparison with competitive baselines on EuroSAT using CLIP ViT-B/16. "Ep" denotes one training epoch. "Per-task" denotes the cumulative cost required to complete training for one incremental task under each method's original training schedule. ZS Overall results are averaged over the 24 zero-shot evaluation datasets.

| Method | Peak GPU | Time/Ep | Time/Task | FLOPs/Ep | FLOPs/Task | CIL Last. ↑ | CIL BWT ↑ | ZS Overall Last. ↑ | ZS Overall SCR ↑ |
|---|---|---|---|---|---|---|---|---|---|
| MoE4Adapter | 4.51G | 15s | 15s | 68.16B | 68.16B | 62.52 | -15.56 | 58.86 | -4.16 |
| DMNSP | 5.13G | 28s | 112s | 71.94B | 287.76B | 69.64 | -3.40 | 58.82 | -2.65 |
| GIFT | 21.43G | 82s | 434.6s | 264.26B | 1400.58B | 69.52 | -2.57 | 59.31 | -1.04 |
| DFA-MoE | 4.74G | 37s | 37s | 162.05B | 162.05B | 72.25 | -2.96 | 60.21 | -0.79 |

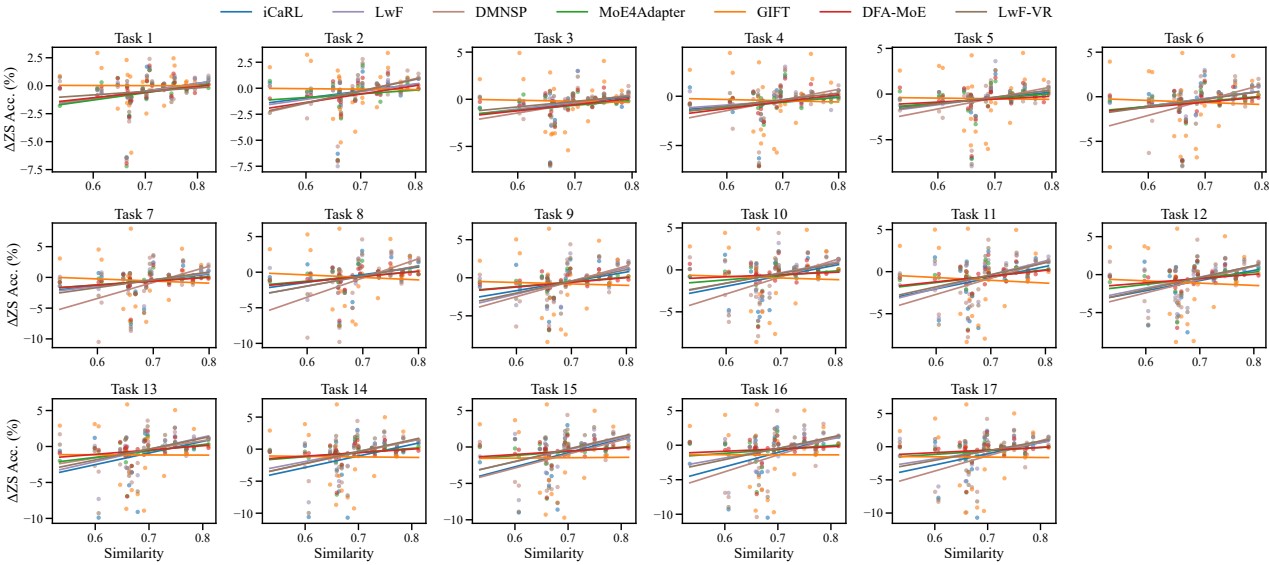

*Figure 10.* Visualizing the confounding effect of positive transfer across all incremental tasks on the **OxfordFlower** dataset. The sub-figures report the change in zero-shot accuracy ($\Delta$ZS Acc.) on each evaluation domain after completing each of the 17 incremental tasks respectively. In each plot, there are 24 points, each representing a specific zero-shot evaluation domain. The x-axis denotes the domain's similarity to the current CIL task, while the y-axis tracks the change in zero-shot accuracy. The colored lines represent the linear regression fits for the specific methods listed in the legend at the top of the figure, highlighting the positive correlation between similarity and zero-shot degradation.

objective. However, this cost is accompanied by substantial improvements in both CIL performance and pre-trained knowledge retention. Compared with DMNSP, DFA-MoE has a slightly higher per-epoch training time, but requires much lower per-task cost under the original training schedules of each method. Specifically, DFA-MoE reduces the training time per task from 112s to 37s and the per-sample FLOPs per task from 287.76B to 162.05B, while achieving better CIL Last Accuracy and ZS Overall SCR. Compared with the replay-based GIFT, DFA-MoE avoids the heavy computational and memory overhead caused by synthetic replay. It uses much less peak GPU memory, training time, and FLOPs, while still achieving stronger CIL performance and better zero-shot retention.

Overall, DFA-MoE provides a favorable trade-off between computational efficiency and dual-forgetting mitigation. Although contrastive alignment increases the per-epoch cost, the method remains efficient at the task level and achieves the best overall balance among the compared methods.

## E. Visualization of Confounding Effect of Positive Transfer on Zero-shot Evaluation

This section provides a comprehensive extension of the analysis presented in Figure 3 of the main paper. We visualize the relationship between the change in zero-shot accuracy ($\Delta$ZS Acc.) on evaluation domains and their similarity to the newly learned incremental task. The following figures present these scatter plots for all incremental tasks on the EuroSAT (Figure 8), OxfordFlower (Figure 10), and FER2013 (Figure 11) datasets. The analysis includes several representative baselines (LwF (Li & Hoiem, 2017), iCaRL (Rebuffi et al., 2017), DMNSP (Kang et al., 2025), GIFT (Wu et al., 2025), LwF-VR (Ding et al., 2022), and MoE4Adapter (Yu et al., 2024a)) and our proposed DFA-MoE. The Chordprompt* method

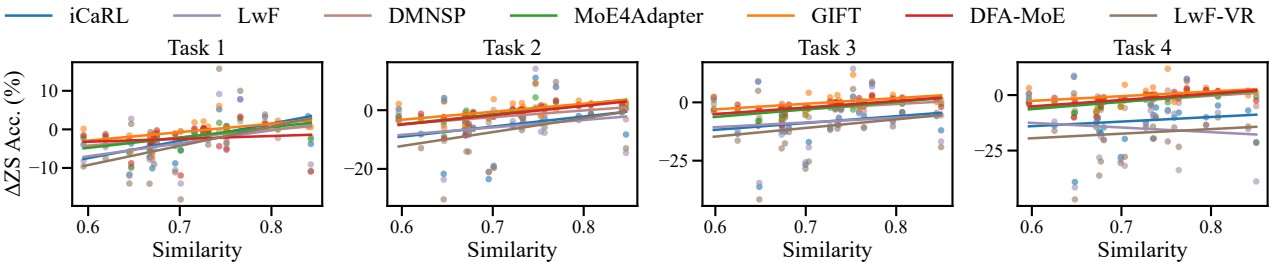

*Figure 11.* Visualizing the confounding effect of positive transfer across all incremental tasks on the **FER2013** dataset. The sub-figures report the change in zero-shot accuracy (ΔZS Acc.) on each evaluation domain after completing each of the 4 incremental tasks respectively. In each plot, there are 24 points, each representing a specific zero-shot evaluation domain. The x-axis denotes the domain's similarity to the current CIL task, while the y-axis tracks the change in zero-shot accuracy. The colored lines represent the linear regression fits for the specific methods listed in the legend at the top of the figure, highlighting the positive correlation between similarity and zero-shot degradation.

is excluded from this specific visualization to maintain clarity. Its zero-shot accuracy suffers from severe degradation, and including these extreme values would disproportionately expand the y-axis scale, thereby obscuring the subtle trends of other methods. This underperformance stems from our necessary adaptation of the method: although the original Chordprompt (Wang & Chen, 2025) operates without explicit task ID, it fundamentally relies on a pre-defined total number of tasks for parameter initialization. This assumption violates the Class-Incremental Learning (CIL) paradigm, where the future task stream is unknown. We remove this dependency to align with the strict CIL paradigm, which inevitably impacted its optimization effectiveness.

A clear and consistent positive correlation is observable across the vast majority of methods and tasks, visualized by the upward-sloping fitted regression lines. This trend provides strong empirical evidence for the confounding influence of positive transfer: higher similarity typically corresponds to less zero-shot degradation, while performance degradation is most severe on domains that are conceptually dissimilar to the current training data. We acknowledge that there are rare exceptions where the correlation appears neutral or slightly negative, such as the MoE4Adapter method in Task 2 of the EuroSAT experiment (see Figure 8). However, these are isolated outliers. The overwhelming consistency of the positive correlation across three distinct datasets and nearly all incremental stages confirms that this tension between local transfer and global retention is a systemic phenomenon in VLM continual learning.

These detailed visualizations further validate the necessity of our SCR, which is explicitly designed to disentangle genuine knowledge preservation from these transfer effects to ensure a rigorous evaluation.

**Discussion: Rationale for Directed Distance.** The observed positive transfer phenomena underscore the rationale for employing the Directed Chamfer Distance ($d_{\text{train}\to\text{eval}}$) to quantify semantic similarity. We posit that a primary driver of positive transfer in this setting is the *partial containment relationship*, where the concepts learned in the current task (subset) represent a specific component of a broader evaluation domain (superset). In such subset-to-superset scenarios, the directed distance from the training data to the evaluation set is naturally minimal, effectively signaling the high relevance of the learned features. In contrast, a symmetric metric (or the reverse direction $d_{\text{eval}\to\text{train}}$) incorporates distances from the numerous non-overlapping "background" classes in the superset. These additional classes act as geometric outliers relative to the training data, *potentially diluting* the similarity score and reducing the metric's sensitivity to the critical subset alignment that drives transfer. Therefore, for the specific purpose of strictly diagnosing transfer-induced gains, the directed metric offers a more targeted and conservative estimation of semantic proximity.

# F. Detailed Performance Trajectories

This section provides a comprehensive, task-by-task visualization of the performance of all evaluated methods on the three main CIL datasets: EuroSAT, OxfordFlower, and FER2013. The following plots in Figure 12, Figure 13 and Figure 14 track the evolution of four key metrics—CIL Last Accuracy, Zero-shot Last Accuracy, Backward Transfer (BWT), and SCR as the models learn new tasks incrementally. These trajectories offer a granular view of how each method handles the dual-forgetting challenge over time.

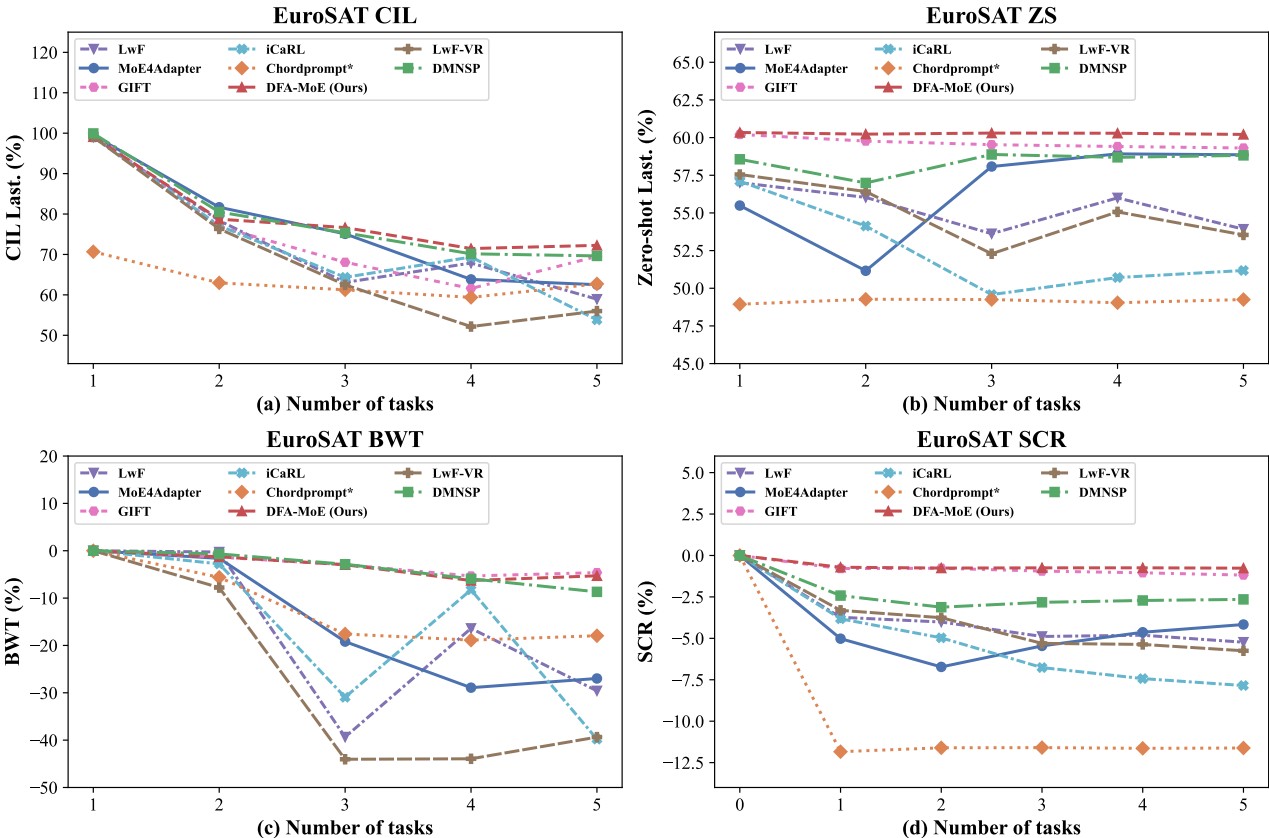

*Figure 12.* Performance trajectories of all methods on the EuroSAT dataset. The plots show the evolution of (a) CIL Last Accuracy, (b) Zero-shot Last Accuracy, (c) BWT, and (d) SCR after the model learns each incremental task.

**Performance on EuroSAT**   Figure 12 illustrates the performance trajectories on the EuroSAT dataset, a task requiring adaptation to the distinct domain of satellite imagery. The results clearly highlight the superior stability of our proposed DFA-MoE, which outperforms the majority of baseline methods. Its Zero-shot Last Accuracy and SCR curves remain consistently high and remarkably flat, closely tracking the zero-shot generalization performance of the original, unmodified CLIP model. This demonstrates a robust and continuous preservation of pre-trained knowledge, a direct result of our dedicated task-agnostic expert designed to prevent foundational knowledge erosion. This stability contrasts sharply with approaches like LwF (Li & Hoiem, 2017), iCaRL (Rebuffi et al., 2017), and MoE4Adapter (Yu et al., 2024a), which exhibit volatility and a clear downward trend.

A nuanced comparison with the top-performing competitors further illuminates the superiority of our design. Regarding DMNSP (Kang et al., 2025), although it achieves a CIL Last Accuracy that tracks closely with ours, the proposed DFA-MoE establishes a clear dominance in all other metrics. We achieve a significantly better BWT, indicating superior stability in learned tasks. The superiority is especially evident in pre-trained knowledge preservation, where DFA-MoE outperforms DMNSP by a large margin in overall performance averaged across all tasks, specifically in SCR (-0.79% vs. -2.65%) and Zero-shot Last Accuracy (60.27% vs. 58.38%). Conversely, while the replay-based GIFT method manages to maintain BWT, Zero-shot Last Accuracy, and SCR comparable to DFA-MoE, it falls short in acquiring new task knowledge. Our method achieves a markedly higher CIL Last Accuracy compared to GIFT (79.51% vs. 75.17%). Taken together, these trajectories confirm that DFA-MoE successfully addresses the critical trade-off between plasticity and stability, achieving a superior and well-balanced approach to mitigating both forms of forgetting where other methods compromise on one front.

**Performance on OxfordFlower**   The performance trajectories on the fine-grained OxfordFlower dataset are presented in Figure 13. On this challenging task, the trend of performance degradation in baseline methods is still pronounced. To enhance visual clarity in the Zero-shot Last Accuracy and SCR plots, the Chordprompt* method has been excluded, as its

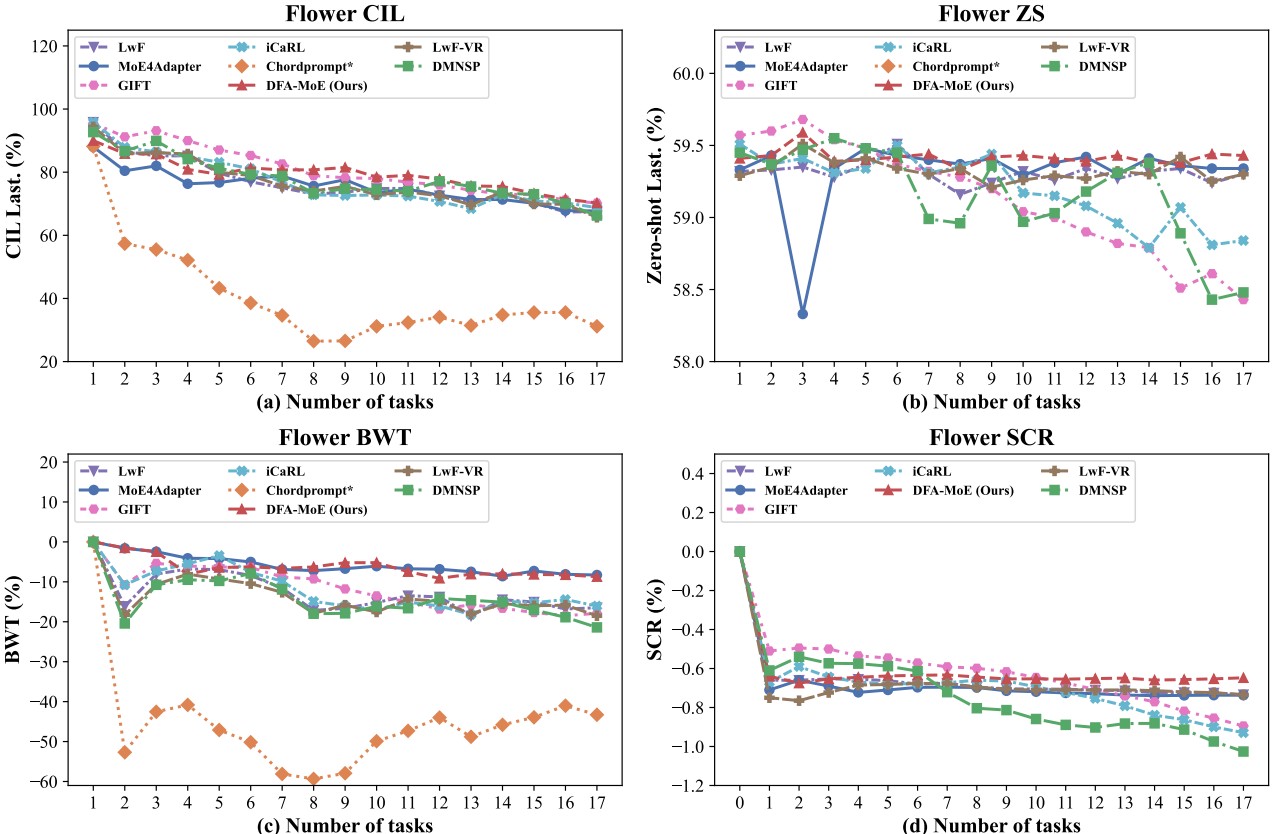

*Figure 13.* Performance trajectories of all methods on the OxfordFlower dataset. The plots show the evolution of (a) CIL Last Accuracy, (b) Zero-shot Last Accuracy, (c) BWT, and (d) SCR after each incremental task.

substantially lower scores would otherwise compress the y-axis and obscure the trends of other methods. As mentioned above, this subpar performance is attributed to the necessary adaptation of the method: the original Chordprompt was not designed for the strict CIL setting and relies on knowing the total number of tasks in advance. To align it with the CIL paradigm, we have to modify the method by removing this dependency, which compromised its optimization effectiveness.

Among the remaining methods, DFA-MoE maintains a clear leadership position, exhibiting remarkably flat and stable curves across all four metrics. A closer inspection of the top-tier competitors reveals the unique comprehensive strength of our approach. For instance, while MoE4Adapter demonstrates strong retention with a marginally superior BWT, DFA-MoE outperforms it across all other dimensions. We maintain a lead in preservation metrics, specifically Zero-shot Last Accuracy and SCR. Most notably, DFA-MoE establishes a substantial advantage in plasticity, surpassing MoE4Adapter by a significant margin in overall CIL Last Accuracy averaged across all tasks (78.86% vs. 75.46%). A different trade-off is observed with GIFT. Although it achieves a slight lead over our method in overall CIL Last Accuracy, DFA-MoE outperforms it in pre-trained knowledge preservation, achieving higher SCR and Zero-shot Last Accuracy. Crucially, our method proves significantly superior in stability, establishing a substantial lead in overall BWT averaged across all tasks (-6.46% vs. -11.56%).

**Performance on FER2013** Figure 14 displays the results on the FER2013 dataset. Adapting to this domain involves a significant shift from natural RGB images to grayscale facial expressions, posing a substantial challenge for maintaining pre-trained generalization. In this setting, DFA-MoE demonstrates a decisive advantage over the majority of baseline methods. When benchmarking against the strongest competitors, DFA-MoE maintains its leadership through a superior balance across all metrics. Specifically, MoE4Adapter emerges as the closest rival, yielding a CIL Last Accuracy comparable to ours. However, DFA-MoE outperforms it across all other metrics. In terms of overall performance averaged across all tasks, our method maintains an edge in Zero-shot Last Accuracy (59.32% vs. 59.07%), SCR (-1.74% vs. -2.00%), and BWT

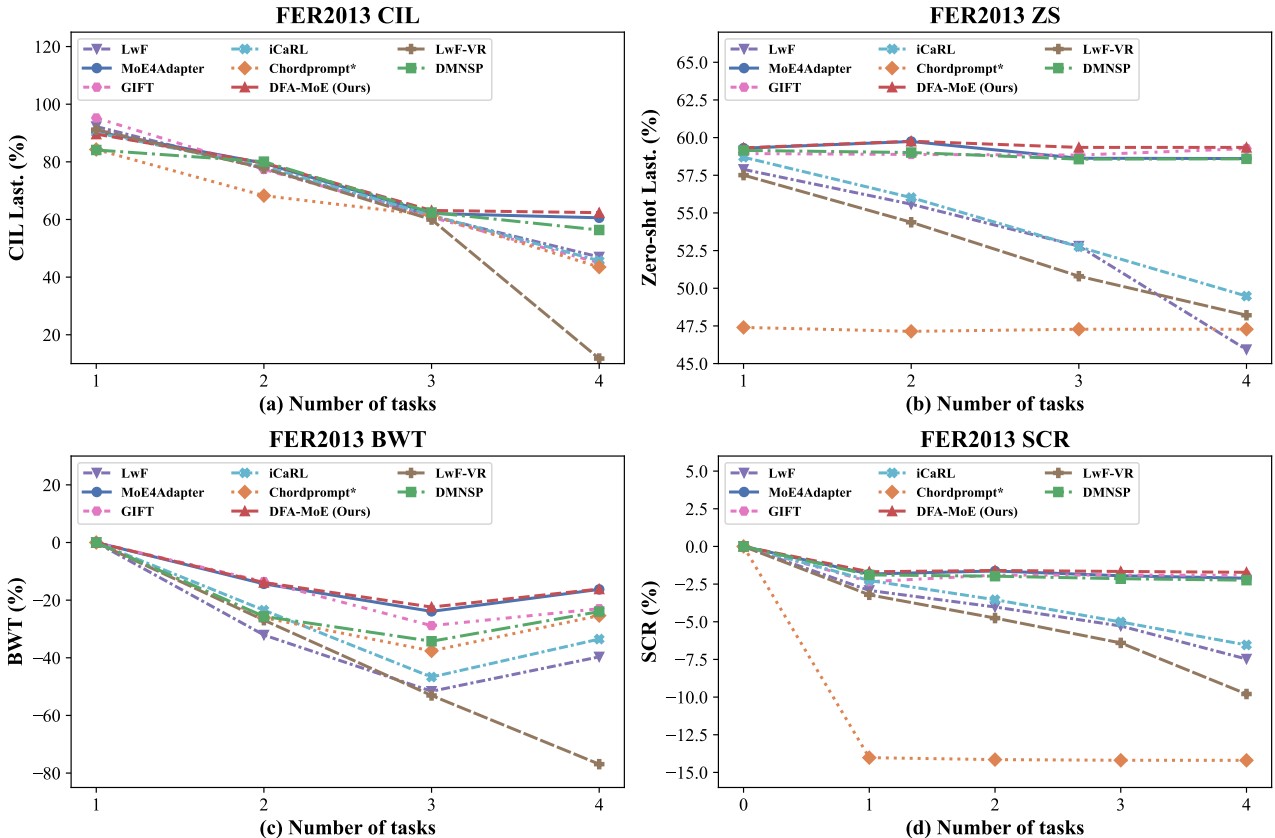

*Figure 14.* Performance trajectories of all methods on the FER2013 dataset. The plots show the evolution of (a) CIL Last Accuracy, (b) Zero-shot Last Accuracy, (c) BWT, and (d) SCR after each incremental task.

(-10.60% vs. -11.13%). Regarding DMNSP, while it matches DFA-MoE in the SCR metric, our method surpasses it with better average accuracies in both CIL Last. (73.18% vs. 70.73%) and Zero-shot Last. (59.32% vs. 58.83%). Crucially, DFA-MoE demonstrates a significantly superior BWT (-10.6% vs. -20.97%), indicating much more stable retention of incrementally learned tasks. Finally, compared to GIFT, DFA-MoE delivers superior results across all evaluated metrics. While GIFT remains competitive in preserving pre-trained knowledge, our method secures a lead in both Zero-shot Last Accuracy and SCR. More importantly, DFA-MoE establishes a significant advantage in incremental learning performance. In terms of overall performance averaged across all tasks, our method outperforms GIFT in both BWT (-10.60% vs. -16.15%) and CIL Last. (73.18% vs. 69.59%). This granular analysis confirms that even among high-performing PEFT and replay strategies, DFA-MoE uniquely optimizes the entire spectrum of evaluation metrics without succumbing to specific weaknesses.

In conclusion, considering the results in Figures 12, 13 and 14 collectively, the proposed DFA-MoE demonstrates superior overall performance among the compared methods.

