# OpenReview forum: "Don't Forget Why You Started: Tackling Dual Forgetting in Vision-Language Continual Learning"
_ICML.cc/2026/Conference — ICML 2026 regular_

### Official Review · Reviewer_yP3h · 2026-03-07

**Soundness:** 3
**Presentation:** 3
**Significance:** 3
**Originality:** 3
**Overall Recommendation:** 4
**Confidence:** 4

**Summary:**

This paper distangles the effect of positive transfer on zero-shot (ZS) performance degradation for continual learning (CL) methods using vision language models (VLMs), proposing a more reliable ZS performance metric. The paper then proposes a method to prevent the ZS performance using parameter efficient expert branch that does contrastive learning on diverse feature queues. Extensive empirical results are provided that show the effectiveness of the proposed method.

**Compliance With Llm Reviewing Policy:**

Affirmed.

**Final Justification:**

My main concerns are adequately addressed in the rebuttal. After considering other reviews and corresponding rebuttals, I want to maintain my original score.

**Key Questions For Authors:**

Please refer to the weakness section above. Regarding how the feature queue is maintained, could authors provide evidence about how many tasks’ features are preserved at a time in the queue?  Why does increasing the queue length have minimal gain in performance?

**Strengths And Weaknesses:**

**Strengths:**
- Although the idea of preserving ZS performance in itself is not novel, the paper proposes a novel angle to show how positive transfer can have a confounding effect on ZS performance evaluation and thus proposes a more balanced and reliable metric for ZS evaluation. This is useful for future research directions of CL.
- Overall, a good paper with a simple method and novel aspects.
- Good empirical evaluation covering CIL evaluation, ablation studies, computational cost analysis, and analysis on queue length.


**Weakness:**
- Although some discussion about existing ZS preservation methods is provided, the empirical comparison with such methods is missing. For instance, ZSCL can be adapted to the setting of this paper. Other ZS preservation methods also need to be discussed and compared with:
  - Gao et al., “Stabilizing Zero-shot prediction: A novel antidote to Forgetting in Continual Vision-Language Tasks” - NeurIPS 2024
  - Liu et al., “C-CLIP: Multimodal Continual Learning for Vision-Language Model” - ICLR 2025

- The proposed evaluation relies on using the existing ZS evaluation dataset, which may not always be available for all models and tasks.
- Minor: The writing can use some improvement, for instance:
  - Lines 174-175: What are the source and target datasets?
  - Line 81: Typo (“homogeneous” is written twice)
  - Line 150-151: What is a task-id-free protocol?
  - Figure 3 description: missing punctuation “.”

---

> ### Author Rebuttal · Authors · 2026-03-31
>
> Thank you for your insightful review. We are encouraged that you found our identification of the “illusion of retention” bias useful and appreciated the simplicity and novelty of our method. (W denotes weakness, Q denotes question.)
>
> **W1: Empirical Comparisons with ZS Preservation Methods**
>
> We deeply appreciate the recommendation. We have added the requested zero-shot preservation methods as baselines, adapted to strict DFA-CIL setting: ZSCL (ICCV 2023) and ZAF (NeurIPS 2024). We could not evaluate C-CLIP (ICLR 2025) as its source code is not publicly available. To ensure fairness, all methods are evaluated using a unified CLIP backbone (ViT-B/16).
>
> | Dataset      | Method   | CIL Last. (↑) | CIL BWT (↑) | ZS Overall Last. (↑) | ZS Overall SCR (↑) |
> | :----------- | :------- | :------------ | :---------- | :------------------- | :----------------- |
> | EuroSAT      | ZSCL     | 54.88         | -15.43      | 60.00                | -1.26              |
> |              | ZAF      | 53.34         | -34.61      | 57.28                | -6.72              |
> |              | **Ours** | **72.25**     | **-2.96**   | **60.21**            | **-0.79**          |
> | OxfordFlower | ZSCL     | 53.34         | -18.78      | 58.55                | -1.37              |
> |              | ZAF      | 69.48         | -17.68      | 59.35                | -0.72              |
> |              | **Ours** | **70.40**     | **-6.37**   | **59.46**            | **-0.66**          |
> | FER2013      | ZSCL     | 55.22         | -21.62      | 56.16                | -3.32              |
> |              | ZAF      | 53.48         | -17.68      | 59.35                | -1.82              |
> |              | **Ours** | **62.37**     | **-13.21**  | **59.38**            | **-1.74**          |
>
> These results demonstrate DFA-MoE’s superior balance. While ZSCL and ZAF attempt to preserve zero-shot capabilities, they struggle significantly with incremental plasticity (CIL Last.) and stability (BWT) in task-id-free settings.
> We will included these baselines and added a dedicated discussion in the revised Related Work section.
>
> **W2: About “The proposed evaluation relies on using the existing ZS evaluation dataset”**
>
> We clarify that the DFA-CIL framework is a dataset-agnostic meta-protocol. Note that our study uses 25 datasets just to provide a rigorous academic benchmark for evaluation and comparison, and the method is highly applicable to specialized domains where standard datasets are unavailable.
>
> Crucially, the SCR metric operates at the concept level via multimodal prototypes $r^i = [v^i; t^i]$ to train the proposed method. This design allows for effective positive transfer calibration using a lightweight proxy pool of diverse concepts, ensuring that foundational stability can be rigorously monitored even with minimal data or within niche domains.
>
> **Q1: Feature Queue Maintenance and Task Representation**
>
> The queue follows a strict FIFO strategy. After each task, we enqueue 32 latent features and dequeue the oldest 32. With a default capacity of $L=128$, the queue dynamically preserves features from exactly the most recent 4 tasks ($128 \div 32 = 4$).
>
> Regarding Figure 7, the saturation of zero-shot performance is consistent with the theoretical finding in [1]: contrastive objectives reach an informative threshold once negative samples cover the underlying latent class distribution. In our single-dataset setting, $L=128$ already provides enough semantic boundaries to anchor the alignment. We conduct an additional supplementary experiment on OxfordFlower confirms this, where increasing $L$ from 128 to 8192 yielded a negligible +0.01 gain. Thus, expanding the queue introduces redundant features without new informational value. We therefore fixed $L=128$ for all 25 datasets (which vary in task counts) to ensure experimental fairness and consistency.
>
> We will update Section 4.2 to clarify the relationship between queue capacity and the architectural of PEFT adapters.
>
> **Minor Writing Improvements**
>
> We humbly accept all suggestions for improving the writing.
>
> *   **Lines 174-175:** Explicitly defined source dataset ($D_t$) as the current training task and target dataset ($D_e$) as the unseen zero-shot evaluation dataset.
> *   **Line 81:** Corrected the "homogeneous" typo.
> *   **Lines 150-151:** Defined "task-id-free protocol" as a setting where no explicit task identifier is provided at inference.
> *   **Figure 3:** Added the missing punctuation in the caption.
>
> **References:**
>
> [1] A Theoretical Analysis of Contrastive Unsupervised Representation Learning ICML 2019.

---

> > ### Author Rebuttal · Reviewer_yP3h · 2026-04-02
> >
> > Thank you for addressing my concerns. My main concern was about the comparison with methods that focus on preserving ZS performance, and the authors provided an adequate response addressing it. Overall, I think the contribution is significant as the paper provides a new angle to distangle positive transfer from ZS retention, proposes a new metric to consider it, and shows clear empirical improvement.
> >
> > After considering other reviews and corresponding rebuttals, I want to maintain my original score. However, some of the issues raised by R-NZ5u do sound concerning (Q1, Q2, Q3), which makes it unclear whether the method is reproducible.

---

> > > ### Author Response · Authors · 2026-04-06
> > >
> > > We sincerely thank you for recognizing the significance of our contributions. Regarding the concerns raised by Reviewer NZ5u (Q1–Q3), we have provided comprehensive justifications. To ensure full transparency and reproducibility, we officially promise to release the complete source code and experimental framework in the camera-ready version.

---

### Official Review · Reviewer_XTzT · 2026-03-08

**Soundness:** 2
**Presentation:** 3
**Significance:** 3
**Originality:** 3
**Overall Recommendation:** 4
**Confidence:** 4

**Summary:**

This paper explores the core challenge of “dual forgetting” faced by visual language models in continual learning: forgetting incremental knowledge of newly acquired classes and forgetting pre-trained knowledge of foundational zero-shot capabilities. To address the issue of forward transfer masking foundational capability degradation in existing evaluation protocols, the authors propose the Dual Forgetting-Aware Continual Learning (DFA-CIL) framework and the Similarity Calibration Retention (SCR) metric. Methodologically, the paper introduces DFA-MoE, a parameter-efficient fine-tuning approach with functional heterogeneity. This method decouples the architecture: an Alignment Pathway anchors the foundational feature space using momentum-enhanced contrastive targets, while a Plasticity Pathway adapts to new tasks through a hybrid of classification and auxiliary contrastive learning.

**Compliance With Llm Reviewing Policy:**

Affirmed.

**Final Justification:**

The authors' rebuttal cleared up most of my confusion. I decided to raise my rating.

**Key Questions For Authors:**

1. Beyond the task-specific experts commonly found in MoE, the proposed DFA-MoE incorporates several carefully designed modules (momentum queue, outer router, inner router, task-agnostic expert). This paper lacks critical ablation studies to analyze the contribution of each module to performance improvement.
2. Does the momentum queue store features extracted from past task samples? The author emphasizes that “the key challenge is the lack of diverse negative samples.” However, since the training data for the experiments all originate from the same dataset, using Momentumqueue yields only limited improvements in diversity compared to directly utilizing samples from the current task—especially when contrasted with the extensive knowledge from pre-training. Figure 7 indicates that increasing the length of the Momentum Queue yields negligible improvement in zero-shot performance.
3. The mechanism by which the inner router obtains the gating vector appears to lack detailed explanation. Based on the diagram, the inner router seems to be untrainable since it does not receive backpropagated gradients (this is my understanding of the dotted line's function in Figure 4). Is the design of the inner router intended to find suitable task-specific experts for the test samples? If so, I'm curious about the accuracy rate of the internal router's task selection on the CIL test set.
4. How exactly are Low-Similarity, Mid-Similarity, and High-Similarity categorized? Is it based on similarity thresholds or rankings? Demonstrating corresponding partitions for different training sets, particularly Task Type, helps validate the rationality of the SCR evaluation criteria. For example, for OxfordFlow, is the similarity higher for fine-grained classification datasets that are also natural RGB images?

**Limitations:**

yes

**Strengths And Weaknesses:**

Strengths:

- The paper suggests that high zero-shot scores in semantically similar domains may merely reflect “forward transfer” within the current task rather than fundamental knowledge retention. This is meaningful for optimizing evaluation criteria for VLM-based continual learning.
- The figures in this paper are clear and easy to read.

Weaknesses:
- The proposed method consists of multiple carefully designed modules, but lacks ablation studies for these modules.
- The SCR metric is heuristic. The paper lacks theoretical or mathematical justification for why this specific decay function can objectively and linearly unravel the confusion effects of positive transfer.
- Some details regarding the experiment setting and methods remain to be supplemented. (See questions for details)

---

> ### Author Rebuttal · Authors · 2026-03-31
>
> We sincerely thank you for your thoughtful review. We address your concerns below (W: Weakness, Q: Question).
>
> **W1: Lack Ablation Studies**
>
> We clarify that a comprehensive ablation study is provided in Table 3 of the original manuscript. We apologize if the naming of these variants caused confusion, and we will refine the labels in the revised manuscript.
>
> **W2: Justification for Disentangle Positive Transfer via SCR**
>
> We clarify that the SCR metric is a theoretically motivated and empirically validated approach to mitigate Positive Transfer (PT) bias, where observed zero-shot performance often conflates foundational stability with task-specific gains. Since geometric dataset distance strongly predicts transferability [1], quantifying the semantic proximity ($d$) between training tasks and evaluation domains is essential to assess PT risk. To transform these distances into actionable weights, we employ $S_k = \exp(-\beta \cdot d)$, a principled mapping that mirrors the t-SNE Gaussian kernel [2] for converting distances into local similarity probabilities.
>
> Our empirical findings (Fig. 3; Appx. C) validate this specific exponential decay, demonstrating a consistent positive correlation between $S_k$ and zero-shot accuracy gains. This confirms $S_k$ captures the magnitude of PT-driven "performance inflation." Consequently, applying the inverse penalty $W_{S_k} = 1 - S_k$ systematically down-weights these transferred domains, disentangling PT bias to rigorously diagnose the VLM's true foundational decay.
>
> **Q1: Contribution of Each Module**
>
> | Variant                                                | CIL Last. | CIL BWT | ZS Overall Last. | ZS Overall SCR |
> | :----------------------------------------------------- | :-------: | :-----: | :--------------: | :------------: |
> | Outer Router (3 $E_s$)                                 |   68.62   | -17.24  |      59.42       |     -2.10      |
> | + Inner Router (3 $E_s$)                               |   69.58   |  -3.30  |      59.70       |     -1.10      |
> | $\rightarrow$ Task-agnostic expert (1 $E_a$ + 2 $E_s$) |   71.74   |  -3.18  |      60.04       |     -0.85      |
> | + Queue (DFA-MoE)                                      |   72.25   |  -2.96  |      60.21       |     -0.79      |
>
> As evidenced by the EuroSAT results, (1) The Inner Router is vital for stability, drastically improving BWT (-17.24 $\to$ -3.30). (2) Replacing one specific expert with a Task-agnostic expert is the primary driver for zero-shot retention, boosting SCR from -1.10 to -0.85. (3) The Momentum Queue provides the final refinement to reach the optimal results across all metrics.
>
> **Q2: Momentum Queue and Negative Sample Diversity**
>
> **Yes**, the momentum queue stores past latent features to maintain class-level diversity. Rather than replicating the pre-training scale, it provides historical negative anchors to prevent feature collapse during continual learning.
>
> Regarding Figure 7, the saturation of zero-shot performance is consistent with the theoretical finding in [3]: contrastive objectives reach an informative threshold once negative samples cover the underlying latent class distribution. In our single-dataset setting, $L=128$ already provides enough semantic boundaries to anchor the alignment. We conducted supplementary experiment on OxfordFlower confirms this, where increasing $L$ from 128 to 8192 yielded a negligible +0.01 gain. Thus, expanding the queue introduces redundant features without new informational value.
>
> We therefore fixed $L=128$ for all datasets (which vary in task counts) to ensure experimental fairness and consistency.
>
> **Q3: Inner Router Mechanism**
>
> We clarify two key aspects of the inner router. First, it is fully trainable via backpropagation; dotted lines in Fig. 4 denote gating weight ($\alpha$) flow rather than gradient blockage, enabling a differentiable Softmax-weighted summation of experts. Second, the router performs soft collaborative fusion rather than hard task identification. By blending representational sub-spaces from a limited expert pool to synthesize optimal features for numerous tasks, the experts function as a shared pool of plasticity rather than discrete task learners. Consequently, "task selection accuracy" is not a relevant metric for this architecture.
>
> **Q4: How exactly are Similarity Bins are categorized**
>
> Categorization is based on rankings of the computed semantic distances relative to the training task. We partition the 24 evaluation datasets into three balanced groups of eight each. For OxfordFlower, its High-Similarity bin clusters other natural RGB datasets like CIFAR100 and Food101. Conversely, the Low-Similarity bin contains domains with modality or color-space shifts, such as SST2 and MNIST.
>
> **References:**
>
> [1] Geometric dataset distances via optimal transport. NeurIPS 2020.
>
> [2] Visualizing data using t-SNE. JMLR 2008.
>
> [3] A Theoretical Analysis of Contrastive Unsupervised Representation Learning. ICML 2019.

---

> > ### Author Rebuttal · Reviewer_XTzT · 2026-04-02
> >
> > Thank you for the response and the additional clarifications and experiments. My concerns are partially addressed.
> >
> > If task-specific experts are shared across all tasks, what does “task-specific” actually mean? I'm curious: for test samples from different task stages, does the inner router's output show a clear task bias? For example, does it tend to assign specific experts to specific tasks, or does it blend experts from different tasks evenly?

---

> > > ### Author Response · Authors · 2026-04-06
> > >
> > > We appreciate your insightful questions. "Task-specific" refers to the experts' functional role in providing plasticity for new knowledge, distinguishing them from the stability-focused "task-agnostic" expert.
> > >
> > > The inner router **does not exhibit task bias**; instead of assigning specific experts to specific tasks, it blends them as a collaborative ensemble to encode incoming information. This balanced utilization is actively enforced by our Noisy Top-K strategy to prevent router collapse and expert starvation (detailed in Appendix B).
> > >
> > > Consequently, the experts act as a unified pool for incremental learning, avoiding the fragmentation of the feature space into isolated, task-bound pathways. Empirical evidence of these stable routing weights is provided at: https://anonymous.4open.science/r/XTzT-1264

---

### Official Review · Reviewer_NZ5u · 2026-03-10

**Soundness:** 2
**Presentation:** 2
**Significance:** 3
**Originality:** 3
**Overall Recommendation:** 3
**Confidence:** 4

**Summary:**

This paper looks at “dual forgetting” in Vision-Language Models (VLMs) like CLIP when they learn new tasks over time. Models struggle in two ways: Incremental Knowledge Forgetting (IKF), where they forget earlier tasks, and Pre-trained Knowledge Forgetting (PKF), where they lose their broad zero-shot abilities. The authors point out a problem with how PKF is usually measured: if new tasks are similar to the evaluation data, the model seems to retain its general knowledge, but that’s actually just positive transfer it’s becoming a narrow specialist without true foundational retention. To fix this, they propose a better evaluation method and a Mixture-of-Experts (MoE) model that separates pathways for stability (keeping old knowledge) and plasticity (learning new tasks).

**Compliance With Llm Reviewing Policy:**

Affirmed.

**Final Justification:**

Given the improvements in Q1 and the load-balancing clarification during the rebuttal phase, I am raising my score from 2 to 3. However, the unresolved concerns around SCR's validity (Q4), which is a core contribution, prevent me from supporting acceptance at this stage. I also hope authors consider adding all these revisions to their manuscripts to strengthen their work and visibility.

**Key Questions For Authors:**

Q1. The performance table reports LwF outperforming iCaRL!!!, which contradicts standard CIL literature. Furthermore, 2025 baselines like Chordprompt report abysmally low scores (~31% vs the authors' ~70%). Can you elaborate on why this is happening?

Q2. In the code, the authors arbitrarily limit Zero-Shot samples to 50 (`router_stats_samples = 50`). This aggressive sub-sampling for evaluation is never justified in the text and could lead to high variance/unreliable Zero-Shot metrics. Why?

Q 3. Which file contains your SCR in the code files? Can you also refer me to the line number in the code?

Q 4. Computing the SCR metric using the original, frozen CLIP backbone ($M_0$) assumes that the original feature space is perfectly isotropic and geometrically flawless. It ignores the reality that semantic boundaries naturally shift as the model specializes, meaning the "objective ruler" is actually biased toward the pre-trained distribution. Can you override my statement about your paper?

I would be willing to increase my score if the authors can clearly address these questions and the weakness section.

**Limitations:**

The hierarchical routing system (Inner and Outer routers) in the Plasticity Pathway is highly susceptible to "routing collapse" (where the network lazily relies on a single expert). The methodology section completely omits any mention of a load-balancing loss, which is mandatory in modern MoE architectures to ensure experts are actually utilized.
Please also refer to the weakness section.

**Strengths And Weaknesses:**

Strenght:
1. Insightful Problem Formulation and Identifying the "Positive Transfer Trap."
2. Extensive Empirical Validation: The author evaluated the framework across a wide array of diverse datasets, demonstrating a robust testing environment.
3. Commitment to reproducibility and open science by providing code.

Major Weaknesses

1. Mathematical Flaws:
   a. Systemic Mathematical Inconsistency: The formalism is sloppy and disconnected. In Equation 1, the authors use $D_A$ and $D_B$ to define Directed Chamfer Distance, but abruptly switch to $D_t$ and $D_e^{(k)}$ in the surrounding text and subsequent equations.

  b. Equation 1 features an unexplained `2` in the denominator ($\frac{1}{2m}$). Standard Chamfer Distance does not use this, and the authors fail to justify it.

  c. Undefined Critical Hyperparameters: The equations rely on black box tuning knobs that are never defined. Equation 2 introduces $\beta$ , and Equation 10 introduces $\lambda$. Without defining these values.


2. My second problem with the paper is clarity in sections 3 and 4. The text structure should start with motivation, then contribution then the key message of the contribution. Additionally, the flow of text is not consistent, for example, 4.1 is "related work".

3. The paper claims to solve Continual Learning without relying on historical data, yet the Alignment Pathway relies heavily on a Momentum Queue of past features to prevent collapse.

4. Scaling Mismatch: Equation 9 ($o_l = \beta_{gen} o_{gen} + \beta_{spec} o_{spec}$) adds outputs from a *frozen* expert and *trainable* experts. Without explicit normalization between these two pathways, the trainable pathway's gradients will likely explode or drown out the frozen pathway over sequential tasks.

5. The two-stage training mechanism (first locking $E_a$, then freezing it to train $E_s$) effectively doubles the forward/backward pass requirements for every new task. The authors present this as "Parameter-Efficient" (PEFT) but hide the massive training-time cost.

6.Waste of Space / Redundant Figures: Figures 1 and 2 are effectively filler. They consume valuable page real estate to illustrate concepts that are fully and better covered by the comprehensive pipeline diagram in Figure 4. I am not insisting on this if you clarify the key message of each figure 1 and 2.

Minor Weaknesses:

1. Careless Repetition: Poor proofreading is evident. For example, line 81 contains the stuttering typo: "Unlike homogeneous homogeneous Mixture-of-Experts..."
2. Abbreviation Mismanagement: The paper is littered with acronym errors.
* "PEFT" is defined redundantly multiple times (e.g., column 1 line 81, and column 2 line 67).
* "VLM" is used on line 29 of the second column without ever being defined.
* Baselines like "GIFT" and "LwF" are dropped into the text and tables without ever being spelled out or properly introduced in the methodology.
3. Over-engineered Notation for Simple Math: Equation 4 uses a convoluted double-summation structure ($\sum \sum$) just to express a standard weighted arithmetic mean.
4. Bibliography Formatting Issues: The reference list is sloppy. Several papers (like *Sun et al. 2025*) are missing publication venues or publishers, and URLs (like the Everingham 2007 citation) are broken or split across lines.

---

> ### Author Rebuttal · Authors · 2026-03-31
>
> We sincerely thank you for your thoughtful review. We address your concerns below. (W: Weakness, Q: Question).
>
> **W1: Clarification on Mathematical Flaws**
>
> **a:** We apologize for the inconsistent notation. We will unify $D_A/D_B$ to $D_t/D_e^{(k)}$ throughout the text.
>
> **b:** The 1/2 factor is a normalization constant. Since the prototype ($r = [v; t]$) sums the squared visual and textual distances, dividing by 2 yields the mean discrepancy across both modalities.
>
> **c:** We apologize for the omission and clarify that $\beta=1.0$ is a temperature scaling factor, while $\lambda=0.001$ (referenced in Section 5.1) is a loss-balancing coefficient.
>
> **W2: Structural Logic of Sections 4.1**
>
> We clarify that Section 4.1 is intended as a Preliminary section providing essential MoE background. In the revised version, we will restructure it.
>
> **W3: Reliance on "Historical Data"**
>
> We do not claim to solve Continual Learning without relying on historical data. As described in Section 4.2, we explicitly utilizes historical latent features via a momentum queue.
>
> **W4: Scaling Mismatch and Gradient Explosion**
>
> We clarify that DFA-MoE indeed incorporates an implicit normalization mechanism. We address scaling mismatch by combining a Softmax-based convex weight scheme ($\beta_{gen} + \beta_{spec} = 1$) with unified zero-initialized experts. Further, we incorporate residual connections to maintain stable gradient flow and prevent explosion.
>
> **W5: Training-time Cost**
>
> As evidenced by the training cost comparison on EuroSAT:
>
> | Method             | Param.  | Peak GPU Usage | Training Time/Epoch | Total Training Time/Task |
> | ------------------ | ------- | -------------- | ------------------- | ------------------------ |
> | MoE4Adapter        | 4.03M   | 4.51G          | 15s                 | 15s (1 Ep)               |
> | DMNSP              | 1.98M   | 5.13G          | 28s                 | 112s (4 Ep)              |
> | GIFT               | 149.62M | 21.43G         | 82s                 | 434.6s (>5 Ep)           |
> | DFA-MoE (proposed) | 1.05M   | 4.74G          | 37s                 | 37s (Stages 1 and 2)     |
>
> _(Note: M=Million, G=GB, s=seconds,)_
>
> While our two-stage design increases per-epoch cost, DFA-MoE reduces total training time per task by converging in just one epoch. We provide a supplemental experiment right here to prove this advantage. If other methods (e.g., DMNSP) use one epoch as our method does, their CIL Last Acc. will decrease. Specifically, DMNSP's CIL Last Acc. falls from 69.64% to 67.89%.
>
> **W6: Space Efficiency and Redundant Figures**
>
> We will consolidate Figures 1 and 2 into a single introductory diagram to save space.
>
> **Q1. Baseline Performance**
>
> **LwF vs. iCaRL:** We clarify that the relative performance between iCaRL and LwF in the VLM is not strictly hierarchical but is highly sensitive to dataset characteristics and task partitioning. For detailed results, please refer to Table 1 at the link provided at the end of this rebuttal.
>
> **Chordprompt:** Following [1], we adapted Chordprompt from its original MTIL setting to a task-id-free CIL protocol. By removing the requirement for total task counts, we ensure a realistic and fair comparison with other methods. The observed performance drop validates our argument: even methods designed for PKF struggle under rigorous CIL conditions.
>
> **Q2: Aggressive Sub-sampling in Zero-Shot Evaluation**
>
> We clarify that `router_stats_samples` is used strictly for debugging. All zero-shot results reported in the paper were computed using the **full test sets**, ensuring that our metrics are reliable and free from any sub-sampling bias.
>
> **Q3: Location of the SCR Implementation**
>
> We apologize for omitting the SCR script. The supplementary material provides the core training code, while the SCR similarity calculation is currently performed offline in a separate repository. We promise to consolidate and open-source the entire pipeline in the final version.
>
> **Q4. Frozen Backbone as a Ruler**
>
> We acknowledge that semantic boundaries shift during specialization, but SCR is not biased by the initial distribution. $M_0$ serves as a fixed initial state to quantify relative performance changes as the model specializes.  Critically, SCR focuses on the relative difference from a constant reference point rather than assuming the reference itself is "geometrically flawless". This allows SCR to objectively monitor foundational stability regardless of the initial model's specific geometry.
>
> **Limitation: Routing Collapse**
>
> We effectively mitigates "routing collapse." by using a noisy top-k gate. Please refer to the Figure1 at the link provided at the end of this rebuttal.
>
> **Minor Weaknesses:** We will implement all requested corrections.
>
> **Link**: https://anonymous.4open.science/r/SCR_and_Router-8577
>
> **Reference:** [1] Boosting continual learning of vision-language models via mixture-of-experts adapters. CVPR2024

---

> > ### Author Rebuttal · Reviewer_NZ5u · 2026-04-02
> >
> > Thank you for your response. I have some strong concerns. I could consider other partially solved.
> >
> > W4 is Unconvincing.
> > My concern was specifically about the interaction between the frozen and trainable pathways over sequential tasks. Saying "we have residual connections" doesn't address whether the trainable pathway's contribution gradually dominates the frozen one across, say, 10 tasks. Additionally, isn't this change affecting the result you reported?
> >
> > W5. How can "Training Time/Epoch" and "Total Training Time/Task" both be 37s if there are two stages? Either each stage is sub-epoch (which is unusual and needs explanation), or the column headers don't map cleanly to a two-stage procedure. The supplemental DMNSP comparison (69.64% → 67.89% at 1 epoch) is a nice touch but a drop of only ~1.7% actually weakens your argument; it suggests DMNSP is reasonably robust to fewer epochs, not that multi-epoch training is wasteful.
> >
> > GIFT epochs: The total training time of 434.6s divided by 82s/epoch yields approximately 5.3 epochs, why is this reported as ">5 Ep" rather than the precise epoch count?
> >
> > GIFT uses 149.62M parameters and 21.43G of GPU memory, which is 1–2 orders of magnitude larger than the other methods. Can youu clarify whether all methods share the same frozen backbone, and whether this comparison is equitable?
> >
> > Q1: It moves from red flag to partially addressed.  I checked the link. The core point, that VLM-based CL doesn't preserve the traditional iCaRL > LwF hierarchy, is plausible and worth stating. But you need error bars, and explain the mechanism rather than just showing that it happens. I could have run your code, but it isn’t runnable..
> >
> > Q2. Claiming router_stats_samples=50 is only for debugging is fine, but it's a bad look to leave debugging artifacts in released code without documentation.
> >
> > Q3. Unconvincing.
> >
> > Q4. Your argument is circular: "SCR measures relative change from a fixed reference, so it doesn't matter if the reference is flawed." But the whole point of the question is that what counts as a meaningful change depends on the geometry of the reference space. If the frozen CLIP space clusters certain semantics artificially, SCR could register "stability" where actual knowledge has degraded, or vice versa.
> >
> >
> > Limitation. "Noisy top-k" is indeed a known mitigation, but it's not a substitute for an explicit load-balancing loss, which is why I specifically asked about. Pointing to a figure showing balanced routing on one dataset doesn't prove robustness.

---

> > > ### Author Response · Authors · 2026-04-06
> > >
> > > We sincerely thank you for your thoughtful review. We address your concerns below.
> > >
> > > **W4**: Trainable pathway dominance is prevented by dual structural constraints: zero‑initialized experts within the residual architecture keep $o_{spec}$ bounded and controllable throughout incremental learning, while a Softmax-based convex combination ($\beta_{gen} + \beta_{spec} = 1$) maintains a strict proportional balance with the frozen foundation. Our empirical results confirm that these routing weights remain stable across all sequential tasks (see link at rebuttal bottom).
> > >
> > > **W5**: DFA-MoE’s 37s/task indeed covers two sequential stages in a single epoch. Our method establishes a superior efficiency-performance frontier: even with 4 epochs, DMNSP cannot match DFA-MoE’s 1-epoch results. Under a matched 1-epoch budget, DMNSP’s SCR degrades to -4.78%, validating DFA-MoE’s optimized balance for dual forgetting within a minimal computational budget.
> > >
> > > | **Method**        | **Total Time/Task** | **CIL Last. (↑)** | **CIL BWT (↑)** | **ZS Overall Last. (↑)** | **ZS Overall SCR (↑)** |
> > > | ----------------- | ------------------- | ----------------- | --------------- | ------------------------ | ---------------------- |
> > > | DMNSP (4 Epochs)  | 112s                | 69.64%            | -3.40%          | 61.50%                   | -3.30%                 |
> > > | DMNSP (1 Epoch)   | 28s                 | 67.89%            | -4.85           | 59.10%                   | -4.78%                 |
> > > | DFA-MoE (1 Epoch) | 37s                 | 72.25%            | -2.96%          | 61.95%                   | -0.39%                 |
> > >
> > > **About "GIFT"**: To ensure fairness, all evaluated methods utilize the same pre-trained CLIP. The significant resource disparity stems from the difference between full-parameter fine-tuning (GIFT) and our PEFT paradigm. This comparison with latest baseline highlights DFA-MoE’s advantage: achieving a superior dual-forgetting balance at a fraction of the hardware and computational costs required by full-tuning baselines.
> > >
> > > **Q1**: We follow the implementation from ZSCL (ICCV 2023), a seminal VLM-CL baseline widely recognized as the evaluation standard, with subsequent works consistently benchmarking against this version.
> > >
> > > Due to the incompatibility between CLIP's native classification logic and iCaRL's traditional NME classifier, the NME classifier cannot be integrated into CLIP. Furthermore, since iCaRL does not store historical text prompts, this adaptation computes distillation logits by contrasting historical images against current text prompts. This mechanism may lead to unstable modality alignment.
> > >
> > > Our 5-seed evaluation on ImageNet-R confirms this instability.
> > >
> > > | **Setting** | **LwF**           | **iCaRL**         | **Leader** |
> > > | ----------- | ----------------- | ----------------- | ---------- |
> > > | B100-10     | 78.81 ± 0.51%     | **79.38 ± 0.47%** | iCaRL      |
> > > | B100-20     | **73.30 ± 0.16%** | 73.03 ± 0.09%     | LwF        |
> > >
> > > **Q2**: We sincerely regret this oversight. We will thoroughly clean the codebase in the final release.
> > >
> > > **Q3**: Due to institutional policies, we cannot share the code via external links.  We then provide the core implementation logic for SCR below:
> > >
> > > ```
> > > def compute_scr(zero_shot_m: list, similarity_m: list) -> list:
> > >     # Zero_shot_m[0] is base.
> > >     base_acc = zero_shot_m[0]
> > >     scores = []
> > >     for idx, similarity in enumerate(similarity_m):
> > > 	    # Weights: max(0, 1-S).
> > >         weights = [max(0.0, 1.0 - s) for s in similarity]
> > >         sum_weights = sum(weights)
> > >         current_acc = zero_shot_m[idx + 1]
> > >         # Acc deltas
> > >         deltas = [float(current_acc[k]) - float(base_acc[k]) for k in range(len(similarity))]
> > >         # Calculate SCR
> > >         if sum_weights > 0:
> > >             scr = sum(d * w for d, w in zip(deltas, weights)) / sum_weights
> > >         else:
> > >             scr = sum(deltas) / len(deltas)
> > >         scores.append(round(scr, 2))
> > >     return scores
> > > ```
> > >
> > > Q4: While $M_0$ geometry defines "meaningful change", our goal is specifically to quantify the relative erosion of a VLM’s original zero-shot capabilities. Using the frozen CLIP backbone is essential for an equitable comparison, as it serves as the universal foundation for current VLM-CL benchmarks. Furthermore, the SCR metric is not strictly tied to a specific backbone and can be evaluated with other appropriate VLMs.
> > >
> > > **Limitation**: Because our MoE is fully activated and employs Softmax routing weights, standard load-balancing losses are mathematically constant with zero gradients, rendering them inapplicable.
> > >
> > > We instead employ a Noisy Top-K mechanism to inject exploration noise, ensuring both experts receive sufficient gradients and diverse utilization. Our updated empirical visualizations across all layers and tasks for EuroSAT, OxfordFlowers, and FER2013 consistently confirm dynamic expert utilization without starvation(see link at rebuttal bottom).
> > >
> > > **Link**: https://anonymous.4open.science/r/NZ5u-9044

---

### Official Review · Reviewer_CUVv · 2026-03-11

**Soundness:** 3
**Presentation:** 3
**Significance:** 3
**Originality:** 3
**Overall Recommendation:** 4
**Confidence:** 4

**Summary:**

This paper addresses the dual forgetting problem in CIL for VLMs. It identifies that models suffer from both Incremental Knowledge Forgetting and Pre-trained Knowledge Forgetting. The authors propose the DFA-CIL evaluation framework with a Similarity-Calibrated Retention metric to adjust for semantic similarity bias. Furthermore, they introduce DFA-MoE, a heterogeneous parameter-efficient fine-tuning framework with decoupled pathways: an alignment expert using momentum-enhanced contrastive learning to stabilize the pre-trained space, and plasticity experts to acquire new knowledge and mitigate retroactive interference.

**Compliance With Llm Reviewing Policy:**

Affirmed.

**Final Justification:**

After reading other reviews and rebuttals, I decide to keep my initial score.

**Key Questions For Authors:**

1. Can you provide direct evidence that "representation deviation from PTM" strictly equals "zero-shot generalization degradation"? Does the zero-shot performance actually collapse without the strict contrastive alignment constraint, or does the feature space merely shift?

2. What is the exact quantitative computational overhead (e.g., wall-clock training time per epoch, FLOPs, and peak GPU memory) of adding the contrastive loss and momentum queue compared to a standard PEFT baseline?

**Limitations:**

see above.

**Strengths And Weaknesses:**

Strengths:

1. Identifying the "illusion of retention" bias caused by domain similarity in current PKF evaluations provides a more rigorous view for VLM adaptation.

2. Logical Decoupled Architecture: Separating the "alignment" and "plasticity" functions into heterogeneous experts logically balances the retention of foundational knowledge with the acquisition of new concepts.

3. The performance improvement is clear.

Weaknesses:

1. The benchmark heavily relies on maintaining alignment with the Pre-Trained Model representation. However, representation shift during CIL is a natural consequence of adaptation. Deviating from the PTM does not inherently equate to a degradation in zero-shot generalization; it might simply be a benign refinement of the feature space. This may heavily challenge the SCR metric to adjust for knowledge retention ability.

2. While categorized as parameter-efficient, the introduction of contrastive learning—especially coupled with a momentum queue—significantly increases computational complexity. The paper lacks a concrete analysis of the additional training time, FLOPs, and peak GPU memory footprint caused by computing massive negative sample pairs during training.

3. The paper broadly uses "Continual Learning," but given the specific formulation, using "Class-Incremental Learning (CIL)" is much more precise and aligned with the actual problem.

---

> ### Author Rebuttal · Authors · 2026-03-31
>
> We sincerely thank you for your thoughtful review. We have carefully considered your comments and hope our responses below provide the necessary clarification. (W denotes weakness, Q denotes question.)
>
> **W1: Robustness of the SCR Metric to Representation Shift**
>
> We clarify that our benchmark does not measure the representation shift from the PTM. It evaluates the deviation in downstream task accuracy from the original zero-shot baseline. Therefore, feature deviation does not inherently incur a penalty in our evaluation. If a representation shift is indeed a benign refinement, the downstream zero-shot accuracy will naturally increase. This will lead to a higher SCR score.
>
> We strictly use the frozen PTM in only two specific ways. First, as defined in Equation 3, it provides this initial baseline to measure the absolute change in zero-shot performance. Second, as detailed in Section 3.2 (Equations 1 and 2), it calculates the semantic similarity between datasets. This similarity determines the evaluation weights to explicitly filter out positive transfer. As stated in our paper, the PTM acts purely as a "static, objective ruler." It is not a training constraint.
>
> In other word, we do not force the updated model to align with the PTM representation and SCR penalizes actual performance loss rather than representation shift.
>
> **W2&Q2: Exact Quantitative Computational Overhead**
>
> To address this comment, we conduct a quantitative overhead analysis using the CLIP backbone on the EuroSAT dataset. The exact computational costs are summarized in the table below:
>
> | Method             | Peak GPU | Training Time/Ep | Per-sample  FLOPs/Ep | CIL Last.(↑) | CIL BWT (↑) | ZS Overall Last. (↑) | ZS Overall   SCR (↑) |
> | :----------------- | :------- | :--------------- | :------------------- | :----------- | :---------- | -------------------- | :------------------- |
> | Vanilla MoE        | 4.51G    | 15s              | 68.16B               | 62.52        | -15.56      | 58.86                | -4.16                |
> | + CA               | 4.69G    | 36s              | 162.05B              | 71.74        | -3.18       | 60.04                | -0.85                |
> | + CA + Q (DFA-MoE) | 4.74G    | 37s              | 162.05B              | 72.25        | -2.96       | 60.21                | -0.79                |
>
> *(Note: Ep=Epoch, G=GB, s=seconds, B=Billion. Overall ZS Results are averaged over 24 datasets.)*
>
> We acknowledge that Contrastive Alignment (CA) introduces a moderate increase in per-epoch training cost (compared with Vanilla MoE); however, this increase can be well justified by the substantial performance gains achieved.
>
> As shown in the table, while the per-sample FLOPs and training time moderately increase, this yields a 10% improvement (62.52 vs. 72.25) in CIL accuracy and a dramatic reduction in forgetting (BWT from -15.56 to -2.96).
>
> Additionally, the momentum queue further boosts performance with negligible additional training time (increasing from 36s to 37s only). Therefore, we acknowledge that total computational footprint for the entire training process remains highly competitive and practical.
>
> **W3: Terminology precision (CL vs. CIL).**
>
> We agree that "Class-Incremental Learning (CIL)" is a more precise term for our problem formulation. We will update the terminology to CIL throughout the revised manuscript to ensure technical precision.
>
> **Q1: Representation Deviation & Zero-shot Performance Collapse**
>
> - Does "representation deviation from PTM" strictly equal "zero-shot generalization degradation"?
>   - No, they are not strictly equivalent. Learning new tasks inherently requires some feature deviation, which causes positive transfer on similar datasets. Figures 3 and 8-10 show that average zero-shot performance improves on datasets similar to the CIL task but declines on those that are semantically distant. This positive transfer inflates average zero-shot accuracy and masks the actual loss of foundational knowledge. Our SCR metric addresses this issue by discounting positive transfer on similar datasets to reveal the true performance drop. In this case, the deviation manifests as a tangible degradation.
>
> - Does ZS performance actually collapse without the strict contrastive alignment constraint, or does the feature space merely shift?
>   - Without the contrastive alignment constraint, the feature space shifts and erodes foundational knowledge. Table 3 in the paper shows that the average zero-shot accuracy drops from 60.04% to 59.70% without alignment. The overall SCR also decreases to -1.11% as expected. This decline is most visible in Low-Similarity domains, as shown in the table. Without the constraint, the Low-Similarity SCR drops significantly from -1.58% to -1.78%. Adding the alignment restores this score to -1.41%. This demonstrates that the feature drift could indeed cause actual knowledge loss and degrade the ZS performance.

---

> > ### Author Rebuttal · Reviewer_CUVv · 2026-04-02
> >
> > Regarding computational overhead, it needs to be compared with other baselines.

---

> > > ### Author Response · Authors · 2026-04-02
> > >
> > > We thank the reviewer for this constructive feedback. Accordingly, we further evaluated and compared the computational overhead with competitive baselines, using the CLIP backbone on the EuroSAT dataset.
> > >
> > > | Method         | Peak GPU | Training Time/Ep | Training Time/Task | Per-sample FLOPs/Ep | Per-sample FLOPs/Task | CIL Last.(↑) | CIL BWT (↑) | ZS Overall Last. (↑) | ZS Overall SCR (↑) |
> > > | :------------- | :------- | :--------------- | :----------------- | :------------------ | :-------------------- | :----------- | :---------- | :------------------- | :----------------- |
> > > | MoE4adapter    | 4.51G    | 15s              | 15s                | 68.16B              | 68.16B                | 62.52        | -15.56      | 58.86                | -4.16              |
> > > | DMNSP          | 5.13G    | 28s              | 112s               | 71.94B              | 287.76B               | 69.64        | -3.40       | 58.82                | -2.65              |
> > > | GIFT           | 21.43G   | 82s              | 434.6s             | 264.26B             | 1400.58B              | 69.52        | -2.57       | 59.31                | -1.04              |
> > > | Ours (DFA-MoE) | 4.74G    | 37s              | 37s                | 162.05B             | 162.05B               | 72.25        | -2.96       | 60.21                | -0.79              |
> > >
> > > *(Note: Ep=Epoch, G=GB, s=seconds, B=Billion. ZS Overall Results are averaged over 24 datasets.  “Per-Task" metrics denote the total cumulative cost (Time or FLOPs) required to complete the entire training process for one incremental task. )*
> > >
> > > While our two-stage design increases the per-epoch cost relative to MoE4adapter, DFA-MoE strikes the best balance between **overall efficiency and final performance**.
> > >
> > > As shown in the table, although DFA-MoE incurs a higher per-epoch overhead than DMNSP (37s vs. 28s), the training unit in continual learning is the incremental task. Since the number of epochs required for convergence varies across methods, **computational costs should be further compared on a per-task basis**.
> > >
> > > Following the original paper settings, DMNSP and GIFT strictly rely on multi-epoch training to achieve optimal performance. This inflates DMNSP's per-task training time from 28s to 112s and total FLOPs from 71.94B to 287.76B. Similarly, GIFT suffers from massive computational bottlenecks.
> > >
> > > Furthermore, we conducted a supplemental experiment constraining DMNSP to a single-epoch setting. Consequently, its CIL Last Accuracy drops from 69.64% to 67.89%, confirming that DFA-MoE is driven by a highly efficient optimization trajectory.

---

### Decision · Program_Chairs · 2026-04-30

**Decision:**

Accept (regular)

**Comment:**

This paper originally had mixed reviews (4,2,4,3), after  the rebuttal the two negative reviews went up (4,3,4,4). In general the reviewers appreciate the insightful discussion on the preservation of ZS performance and the confounding effect positive transfer can have, as well as the newly proposed SCR metric (the AC agrees this to be a valuable contribution). Among the initial weaknesses the  were computational overhead, lack of theoretical justification, whether deviation of PTM equates degradation, missing explanations (parameter definitions), unclear experimental setup. The rebuttal and the active discussion, let two positive recommendations from 3 reviewers.

For the remaining critical reviewer (NZ5u) most questions were addressed. However, the reviewer was still unsatisfied with respect to the response to (Q4) regarding the role of the pretrained model as an 'objective ruler' (ground-truth model), notign that it is biased towards the pretrained distribution. The AC agrees that the raised point of the reviewer is valid, and should be discussed. Nevertheless, the AC maintains that the proposed metric and the paper make a meaningful contribution in how to measure forgetting from pretrained models. In many scenarios where the pretrained distribution is considerably larger than the target distribution, considering the pretrained model as ground-truth model can be justified.

As a consequence the AC would recommend acceptance for this paper. The authors should prepare a final version incorporating the rebuttal discussion and the newly presented results.